# Role of Momentum in Smoothing Objective Function and Generalizability of Deep Neural Networks

## Abstract

For nonconvex objective functions, including deep neural networks, stochastic gradient descent (SGD) with momentum has faster convergence and better generalizability than SGD without momentum, but a theoretical explanation for this is lacking. Adding momentum is thought to reduce stochastic noise, but several studies have argued that stochastic noise actually contributes to the generalizability of the model, which raises a contradiction. We show that the stochastic noise in SGD with momentum smoothes the objective function, the degree of which is determined by the learning rate, the batch size, the momentum factor, the variance of the stochastic gradient, and the upper bound of the gradient norm. By numerically deriving the stochastic noise level in SGD with and without momentum, we provide theoretical findings that help explain the training dynamics of SGD with momentum, which were not explained by previous studies on convergence and stability, and that resolve the contradiction. We also provide experimental results for an image classification task using ResNets that support our assertion that model generalizability depends on the stochastic noise level.

## 1 Introduction

### 1.1 Background

First-order optimizers that use mini-batch stochastic gradients, such as stochastic gradient descent (SGD) (Robbins & Monro, 1951), SGD with momentum (Polyak, 1964; Rumelhart et al., 1986), and adaptive methods (Duchi et al., 2010; Kingma & Ba, 2015), are the most commonly used methods for solving empirical risk minimization problems that appear in machine learning. These methods have been well studied for their convergence (Bottou et al., 2018; Chen et al., 2021; 2019; Fehrman et al., 2020; Iiduka, 2022a; Loizou et al., 2021; Scaman & Malherbe, 2020; Zaheer et al., 2018; Zhou et al., 2020a; Zou et al., 2019) and stability (Hardt et al., 2016; He et al., 2019; Lin et al., 2016; Mou et al., 2018), and it has been shown that tuning the hyperparameters such as the learning rate, batch size, and momentum factor is essential for successful training. This paper focuses on the SGD with momentum method and provides new insights into the role of the momentum factor.

For nonconvex objective functions, including deep neural networks (DNNs), SGD with momentum experimentally has better generalizability than SGD without momentum (simply "SGD" hereafter), but theoretical explanations for this characteristic have not yet been provided. The generalizability of SGD with momentum has been well studied, and various experimental findings have been reported. While it has been suggested that momentum plays a role in reducing stochastic noise (Defazio, 2020; Cutkosky & Mehta, 2020), stochastic noise has been shown to increase generalizability (Li et al., 2019; Wen et al., 2020; HaoChen et al., 2021), and it has been claimed that stochastic noise can help an algorithm escape from local solutions with poor generalizability (Ge et al., 2015; Jin et al., 2017; Daneshmand et al., 2018; Harshvardhan & Stich, 2021; Kleinberg et al., 2018). Furthermore, several studies (Shallue et al., 2019; Jelassi & Li, 2022; Kunstner et al., 2023) have shown that the gap in convergence speed and generalizability between SGD and SGD with momentum is more pronounced for large batches. There is an inconsistency in that adding momentum should reduce stochastic noise, but because momentum has excellent generalizability, it should have sufficiently large noise, and this contradiction makes it difficult to understand the effect of momentum in DNNs.

The geometry of loss landscapes, in particular the relationship between the flatness of the minima and generalization, has been extensively studied from both theoretical and empirical perspectives (Hochreiter & Schmidhuber, 1997; Keskar et al., 2017; Dziugaite & Roy, 2017; Jiang et al., 2020; Foret et al., 2021). In general, a local optimal solution with flatter neighborhoods is considered to have better generalizability than that with steeper neighborhoods. Several previous studies (Keskar et al., 2017; Liang et al., 2019; Tsuzuku et al., 2020; Petzka et al., 2021; Kwon et al., 2021) developed measures for the flatness of the minima, "sharpness". It has been experimentally observed to correlate with the generalization performance of a model. One previous study (Kleinberg et al., 2018) suggested that the objective function is smoothed by stochastic noise in the optimizer. A more recent study demonstrated that stochastic noise in SGD implicitly smoothes the objective function, that the degree of smoothing caused by stochastic noise in SGD and sharpness both represent the flatness/sharpness of the function, and that the degree of smoothing is correlated with generalization performance (Sato & Iiduka, 2023b). Based on these studies, our study focused on the smoothing of the objective function by stochastic noise in SGD with momentum and the relationship between the degree of smoothing and the generalizability of the model. When considering stochastic noise in optimizers, most previous studies (Zhang et al., 2020; Zhou et al., 2020b; Kunstner et al., 2023) defined stochastic noise as the difference between the mini-batch stochastic gradient and the full gradient. We call this difference "gradient noise." Here, in order to discuss smoothing with stochastic noise, we define optimizer's stochastic noise as the difference between the search direction of the optimizer and the steepest descent direction, which we call "search direction noise." Search direction noise can be viewed as an extension of gradient noise. Note that gradient noise and search direction noise in SGD are consistent with each other.

The simplest method for adding a momentum term to SGD is the stochastic heavy ball (SHB) method (Algorithm 1) (Polyak, 1964). Although it has been widely used in experiments, it is lacking in theoretical analysis. In contrast, the normalized-SHB (NSHB) method (Algorithm 2 with $\nu = 1$) (Gupal & Bazhenov, 1972) has been well analyzed theoretically for convergence and stability but has rarely been used in experiments. Note that the algorithm referred to as "SGD with momentum (SGDM)" in many previous studies is actually NSHB, while that provided by PyTorch (Paszke et al., 2019) and TensorFlow (Abadi et al., 2016) is SHB. Many variants of the momentum method have been proposed, including Nesterov's accelerated gradient method (Nesterov, 1983; 2004; 2013; Sutskever et al., 2013), synthesized Nesterov variants (Lessard et al., 2016), Triple Momentum (Scoy et al., 2018), Robust Momentum (Cyrus et al., 2018), PID control-based methods (An et al., 2018), accelerated SGD (Jain et al., 2018; Kidambi et al., 2018; Varre & Flammarion, 2022; Li et al., 2024), and quasi-hyperbolic momentum (QHM, Algorithm 2) (Ma & Yarats, 2019). This paper focuses on SHB and QHM, which covers many momentum methods, especially NSHB, but does not cover SHB.

**Motivation.** Our main goal in this paper is to resolve the contradiction described in Section 1.1 that exists between momentum and stochastic noise and to clarify the role of momentum in DNNs training. It was recently found that the stochastic noise in SGD implicitly smoothes the objective function and that the degree of smoothing is determined by the learning rate, batch size, and variance of the stochastic gradient (Sato & Iiduka, 2023b). We extend this analysis to SGD with momentum, and by focusing on the stochastic noise between the search direction and the steepest descent direction, we attempt to reveal how momentum is involved in the smoothing of the objective function.

## 1.2 CONTRIBUTIONS

**SGD with momentum's smoothing property (Section 3).** We show that SGD with momentum's search direction noise has a smoothing effect on the objective function, the degree of which is determined by the momentum factor $\beta$, the variance of the stochastic gradient $C_{\text{opt}}^2$, and the upper bound of the gradient norm $K_{\text{opt}}^2$, in addition to learning rate $\eta$ and batch size $b$:

$$\delta^{\text{SGD}} = \eta\sqrt{\frac{C_{\text{SGD}}^2}{b}}, \ \delta^{\text{SHB}} = \eta\sqrt{\left(1 + \hat{\beta}\right)\frac{C_{\text{SHB}}^2}{b} + \hat{\beta}K_{\text{SHB}}^2}, \ \delta^{\text{NSHB}} = \eta\sqrt{\frac{1}{1-\beta}\frac{C_{\text{NSHB}}^2}{b}}, \quad (1)$$

where $\hat{\beta} := \frac{\beta(\beta^2 - \beta + 1)}{(1-\beta)^2}$. (See Assumption 2.1 for exact definitions of $C_{\text{opt}}^2$ and $K_{\text{opt}}^2$). We call this the "degree of smoothing" and denote it by $\delta^{\text{opt}}$ for each optimizer (The subscript "opt" indicates the optimizer's name). The larger the degree, the smoother the function and the greater the dif-

ference from the original function, and the smaller the degree, the smaller the difference between the smoothed function and the original function and the less smoothed the function. Equation (1) shows that a large learning rate and/or small batch size, and/or a large momentum factor, smooths the function even more. Although their results were derived from a different perspective, Smith et al. obtained similar results (Smith et al., 2018). Furthermore, the results of several experimental studies suggest that these hyperparameters are interrelated (Kidambi et al., 2018; Leclerc & Madry, 2020; Kunstner et al., 2023; Fu et al., 2023). Equation (1) also shows that these hyperparameters are interrelated through the degree of smoothing. Therefore, our results provide theoretical support for these previous findings and new insights into the role of hyperparameters in DNNs training.

**Estimation of critical batch size and variance of stochastic gradient (Section 4).** To estimate the variance of stochastic gradient $C_{\mathrm{opt}}^2$ contained in the degree of smoothing as in (1), we consider a critical batch size that is defined by a global minimizer of the stochastic gradient computation cost. We show the existence of a critical batch size in the training of a DNNs with SGD and SGD with momentum and provide a formula for estimating the size. We also estimate the variance of the stochastic gradient for an optimizer from the experimentally estimated critical batch size and show that SGD with momentum, especially SHB, has a smaller variance than SGD. This is the first paper to provide a formula for estimating the critical batch size for SGD and SGD with momentum, and, to the best of our knowledge, the first attempt to estimate the variance of stochastic gradients.

**Why and when momentum improves generalizability (Section 5).** Using the estimated variance of the stochastic gradient, we numerically derived the degree of smoothing introduced by search direction noise (see Figure 1 (Left)). Figure 1 shows that SHB always has a greater degree of smoothing than SGD. Of particular note is that the degree of smoothing depends on the batch size, so that as the batch size increases, the degree of smoothing for SGD and NSHB approaches zero, whereas that for SHB does not decrease thanks to a term independent of batch size (see (1)). Figure 1 also shows that the degree of smoothing introduced by search direction noise is closely related to the generalizability of the model. We observed that an appropriate degree of smoothing, neither too large nor too small, leads to high generalizability. Therefore, the theoretical reason for the phenomenon observed experimentally in some previous studies (Kunstner et al., 2023; Shallue et al., 2019; Jelassi & Li, 2022) that the generalization performance of SHB compared with that of SGD does not deteriorate with an increase in the batch size is that SHB is able to maintain a reasonably large degree of smoothing when the batch is large. Conversely, when the batch is small, the degree of smoothing of SHB is too large, and generalization performance is not excellent. Therefore, the role of the momentum factor in SHB is maintaining a high degree of smoothing even when the batch is large. Furthermore, since an appropriate degree of smoothing leads to high generalizability, we can say that our results are useful for selecting appropriate hyperparameters.

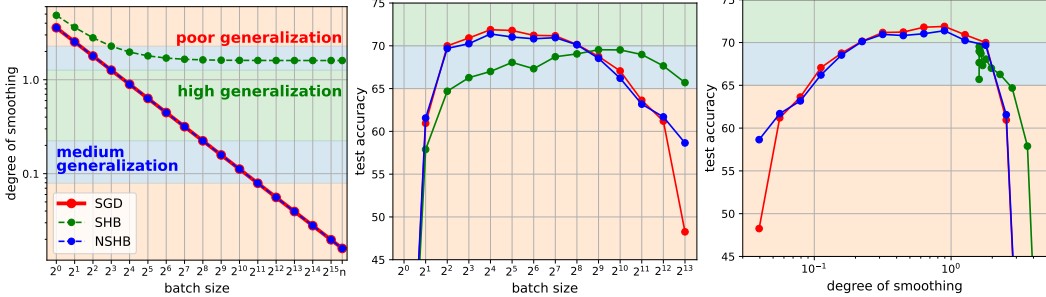

Figure 1: **Left:** Degree of smoothing introduced by search direction noise with $\eta = 0.1$ and $\beta = 0.9$ versus batch size for each optimizer. **Center:** Test accuracy for each optimizer versus batch size. **Right:** Test accuracy for each optimizer versus degree of smoothing in training ResNet18 on CIFAR100 dataset. There is a clear relationship between the degree of smoothing and generalizability; i.e., generalizability is clearly a concave function with respect to the degree of smoothing. Thus, a degree of smoothing that is neither too large nor too small leads to high generalizability. In particular, the degree of smoothing of SHB is not smaller than that of SGD and NSHB when the batch size is large, so the generalizability of SHB remains high even when the batch size is large.

**Resolving the contradiction that exists between momentum and stochastic noise (Section 5).** In summary, adding momentum reduces the gradient noise. Conversely, adding momentum increases

search direction noise, which contributes to smoothing of the objective function. Furthermore, the degree of smoothing can be expressed in terms of hyperparameters including the momentum factor. Since the degree of smoothing is well correlated with the generalizability of the model, the stochastic noise that contributes to the generalizability of the model is search direction noise, not gradient noise. The degree of smoothing of SHB leads to high generalizability because the momentum factor enables it to maintain an appropriate value even when the batch is large. Therefore, the arguments "adding momentum should reduce stochastic noise" and "stochastic noise leads to generalizability" do not conflict, and the contradiction is resolved, namely, "adding momentum reduce gradient noise" and "search direction noise leads to generalizability."

## 2 PRELIMINARIES

### 2.1 NOTATION, DEFINITIONS, AND ASSUMPTIONS

Let $\mathbb{N}$ be the set of non-negative integers. For $m \in \mathbb{N}\backslash\{0\}$, define $[m] := \{1, 2, \ldots, m\}$. $\mathbb{R}^d$ is a $d$-dimensional Euclidean space with inner product $\langle \cdot, \cdot \rangle$, which induces the norm $\| \cdot \|$. $I_d$ denotes a $d \times d$ identity matrix. Let $\mathcal{N}(\boldsymbol{\mu}; \Sigma)$ be a $d$-dimensional normal distribution with mean $\boldsymbol{\mu} \in \mathbb{R}^d$ and variance $\Sigma \in \mathbb{R}^{d \times d}$. The DNNs is parametrized with parameter $\boldsymbol{x} \in \mathbb{R}^d$, which is optimized by minimizing empirical loss function $f(\boldsymbol{x}) := \frac{1}{n} \sum_{i \in [n]} f_i(\boldsymbol{x})$, where $f_i(\boldsymbol{x})$ is a loss function for $\boldsymbol{x} \in \mathbb{R}^d$ and the $i$-th training data point $\boldsymbol{z}_i$ ($i \in [n]$). Let $\xi$ be a random variable that does not depend on $\boldsymbol{x} \in \mathbb{R}^d$, and $\mathbb{E}_\xi[X]$ means the expectation with respect to $\xi$ of a random variable $X$. $\xi_{t,i}$ is a random variable generated from the $i$-th sampling at time $t$, and $\boldsymbol{\xi}_t := (\xi_{t,1}, \xi_{t,2}, \ldots, \xi_{t,b})$ is independent of $(\boldsymbol{x}_k)_{k=0}^t \subset \mathbb{R}^d$, where $b$ ($\leq n$) is the batch size. From the independence of $\boldsymbol{\xi}_0, \boldsymbol{\xi}_1, \ldots$, we can define the total expectation $\mathbb{E}$ by $\mathbb{E} = \mathbb{E}_{\boldsymbol{\xi}_0} \mathbb{E}_{\boldsymbol{\xi}_1} \cdots \mathbb{E}_{\boldsymbol{\xi}_t}$. Let $\mathsf{G}_{\boldsymbol{\xi}_t}(\boldsymbol{x})$ be the stochastic gradient of $f(\cdot)$ at $\boldsymbol{x} \in \mathbb{R}^d$. $\mathcal{S}_t$ is the mini-batch of $b$ samples at time $t$, and $\nabla f_{\mathcal{S}_t}(\boldsymbol{x}_t)$ is the mini-batch stochastic gradient of $f(\boldsymbol{x}_t)$ for $\mathcal{S}_t$; i.e., $\nabla f_{\mathcal{S}_t}(\boldsymbol{x}_t) := \frac{1}{b} \sum_{i \in [b]} \mathsf{G}_{\xi_{t,i}}(\boldsymbol{x}_t)$.

In general, smoothing of a function is achieved by convolving the function with a random variable that follows a normal distribution (Wu, 1996):

**Definition 2.1** (Smoothed function). *Given a function $f \colon \mathbb{R}^d \to \mathbb{R}$, define $\hat{f}_\delta \colon \mathbb{R}^d \to \mathbb{R}$ to be the function obtained by smoothing $f$ as $\hat{f}_\delta(\boldsymbol{x}) := \mathbb{E}_{\boldsymbol{u} \sim \mathcal{N}\left(\boldsymbol{0}; \frac{1}{\sqrt{d}} I_d\right)} \left[ f(\boldsymbol{x} - \delta \boldsymbol{u}) \right]$, where $\delta > 0$ represents the degree of smoothing and $\boldsymbol{u}$ is a random variable from a normal distribution.*

The following lemma represents an important property of smoothed function $\hat{f}_\delta$. This is general and has already been reported by (Hazan et al., 2016). The proof of Lemma 2.1 is in Appendix C.1.

**Lemma 2.1.** *Let $\hat{f}_\delta$ be the smoothed version of $f$; then, for all $\boldsymbol{x} \in \mathbb{R}^d$, $|\hat{f}_\delta(\boldsymbol{x}) - f(\boldsymbol{x})| \leq \mathbb{E}_{\boldsymbol{u}}[\|\boldsymbol{u}\|] \delta L_f$.*

Considering that a local optimal solution with a flatter landscape in the neighborhood yields better generalizability, we can say that the degree of smoothing $\delta$ must be sufficiently large. However, Lemma 2.1 implies that the greater the $\delta$, the greater the gap between original function $f(\boldsymbol{x})$ and smoothed function $\hat{f}_\delta$. Therefore, if the degree of smoothing is constant throughout the training, we can say that its level must be neither too large nor too small.

**Assumption 2.1.** *(A1) $f_i \colon \mathbb{R}^d \to \mathbb{R}$ ($i \in [n]$) is continuously differentiable and a $L_f$-Lipschitz function; i.e., for all $\boldsymbol{x}, \boldsymbol{y} \in \mathbb{R}$, $|f(\boldsymbol{x}) - f(\boldsymbol{y})| \leq L_f \|\boldsymbol{x} - \boldsymbol{y}\|$. (A2) $(\boldsymbol{x}_t)_{t \in \mathbb{N}} \subset \mathbb{R}^d$ is a sequence generated by an optimizer. (i) For each iteration $t$, $\mathbb{E}_{\boldsymbol{\xi}_t}[\mathsf{G}_{\boldsymbol{\xi}_t}(\boldsymbol{x}_t)] = \nabla f(\boldsymbol{x}_t)$. (ii) There exists a non-negative constant $C_{\mathrm{opt}}^2$ for an optimizer such that $\mathbb{E}_{\boldsymbol{\xi}_t}\left[\|\mathsf{G}_{\boldsymbol{\xi}_t}(\boldsymbol{x}_t) - \nabla f(\boldsymbol{x}_t)\|^2\right] \leq C_{\mathrm{opt}}^2$. (A3) For each iteration $t$, the optimizer samples a mini-batch $\mathcal{S}_t \subset \mathcal{S}$ and estimates the full gradient $\nabla f$ as $\nabla f_{\mathcal{S}_t}(\boldsymbol{x}_t) := \frac{1}{b} \sum_{i \in [b]} \mathsf{G}_{\xi_{t,i}}(\boldsymbol{x}_t) = \frac{1}{b} \sum_{\{i \colon \boldsymbol{z}_i \in \mathcal{S}_t\}} \nabla f_i(\boldsymbol{x}_t)$. (A4) There exists a positive constant $K_{\mathrm{opt}}$ for an optimizer, for all $t \in \mathbb{N}$, $\mathbb{E}\left[\|\nabla f(\boldsymbol{x}_t)\|^2\right] \leq K_{\mathrm{opt}}^2$.*

The variance of the stochastic gradient and the upper bound of the gradient are often assumed to be constant for any optimizer, but we define them as $C_{\mathrm{opt}}^2$ and $K_{\mathrm{opt}}$ for each optimizer. The subscript "opt" indicates the optimizer's name. Thus, for example, Assumption (A2)(ii) means that when a sequence $(\boldsymbol{x}_t)_{t \in \mathbb{N}}$ is generated by SGD, there exists $C_{\mathrm{SGD}}^2$ satisfying $\mathbb{E}_{\boldsymbol{\xi}_t}\left[\|\mathsf{G}_{\boldsymbol{\xi}_t}(\boldsymbol{x}_t) - \nabla f(\boldsymbol{x}_t)\|^2\right] \leq C_{\mathrm{SGD}}^2$. Here, $C_{\mathrm{opt}}^2$ depends not only on random variable $\boldsymbol{\xi}_t$ but also on parameter $\boldsymbol{x}_t$. Since different

optimizers yield different $\boldsymbol{x}_t$ at given time $t$, $C_{\mathrm{opt}}^2$ depends on the optimizer, so Assumption (A2)(ii) is valid. Experimental results supporting this assertion are plotted in Figure 5 in Appendix A.1.

## 2.2 ALGORITHMS

We consider two algorithms that are a type of SGD with momentum.

| **Algorithm 1** Stochastic Heavy Ball (SHB) | **Algorithm 2** Quasi-Hyperbolic Momentum (QHM) |
|---|---|
| **Require:** $\boldsymbol{x}_0, \eta > 0, \beta \in [0, 1), \boldsymbol{m}_{-1} := \boldsymbol{0}$ | **Require:** $\boldsymbol{x}_0, \eta > 0, \nu, \beta \in [0, 1), \boldsymbol{d}_{-1} := \boldsymbol{0}$ |
|   **for** $t = 0$ to $T - 1$ **do** |   **for** $t = 0$ to $T - 1$ **do** |
|     $\boldsymbol{m}_t := \nabla f_{\mathcal{S}_t}(\boldsymbol{x}_t) + \beta \boldsymbol{m}_{t-1}$ |     $\boldsymbol{d}_t := (1 - \nu\beta)\nabla f_{\mathcal{S}_t}(\boldsymbol{x}_t) + \nu\beta \boldsymbol{d}_{t-1}$ |
|     $\boldsymbol{x}_{t+1} := \boldsymbol{x}_t - \eta \boldsymbol{m}_t$ |     $\boldsymbol{x}_{t+1} := \boldsymbol{x}_t - \eta \boldsymbol{d}_t$ |
|   **end for** |   **end for** |
|   **return** $\boldsymbol{x}_T$ |   **return** $\boldsymbol{x}_T$ |

In accordance with Gitman et al. (Gitman et al., 2019), we refer to Algorithm 1 as the SHB method. In Algorithm 2, $\nu$ is the coefficient balancing SGD ($\nu = 0$) with NSHB ($\nu = 1$).

# 3 SGD WITH MOMENTUM'S STOCHASTIC NOISE AND SMOOTHING

Kleinberg et al. suggested that stochastic noise in SGD may smooth the objective function (Kleinberg et al., 2018). Sato and Iiduka supported this theoretically and showed that the degree of smoothing is determined by hyperparameters (Sato & Iiduka, 2023b). In this section, we extend this discussion to SHB and QHM.

At time $t$, let $\boldsymbol{\omega}_t^{\mathrm{SHB}}$ be the difference between the search direction of the gradient descent and the search direction of SHB, and let $\boldsymbol{\omega}_t^{\mathrm{QHM}}$ be the difference between the search direction of the gradient descent and the search direction of QHM:

$$\boldsymbol{\omega}_t^{\mathrm{SHB}} := \boldsymbol{m}_t - \nabla f(\boldsymbol{x}_t) \ \text{ and } \ \boldsymbol{\omega}_t^{\mathrm{QHM}} := \boldsymbol{d}_t - \nabla f(\boldsymbol{x}_t).$$

$\boldsymbol{\omega}_t^{\mathrm{SHB}}$ and $\boldsymbol{\omega}_t^{\mathrm{QHM}}$ are search direction noise; it is analogous to "search direction" in the optimization field. Indeed, they take into account not only its direction but also its magnitude. Then, the following theorem holds:

**Theorem 3.1.** *Suppose that Assumptions (A2)(ii), (A3), and (A4) hold, then, for all $t \in \mathbb{N}$,*

$$\mathbb{E}\left[\left\|\boldsymbol{\omega}_t^{\mathrm{SHB}}\right\|\right] \leq \sqrt{\frac{C_{\mathrm{SHB}}^2}{b} + \frac{\beta(\beta^2 - \beta + 1)}{(1 - \beta)^2}\left(\frac{C_{\mathrm{SHB}}^2}{b} + K_{\mathrm{SHB}}^2\right)}, \ \ \mathbb{E}\left[\left\|\boldsymbol{\omega}_t^{\mathrm{QHM}}\right\|\right] \leq \sqrt{\frac{1}{1 - \nu\beta}\frac{C_{\mathrm{QHM}}^2}{b}}.$$

Hence, search direction noise $\boldsymbol{\omega}_t^{\mathrm{SHB}}$ can be expressed as

$$\boldsymbol{\omega}_t^{\mathrm{SHB}} = \sqrt{\left(1 + \frac{\beta(\beta^2 - \beta + 1)}{(1 - \beta)^2}\right)\frac{C_{\mathrm{SHB}}^2}{b} + \frac{\beta(\beta^2 - \beta + 1)}{(1 - \beta)^2}K_{\mathrm{SHB}}^2}\,\boldsymbol{u}_t =: \psi^{\mathrm{SHB}}\boldsymbol{u}_t,$$

where $\boldsymbol{u}_t \sim \mathcal{N}\left(\boldsymbol{0}; \frac{1}{\sqrt{d}}I_d\right)$. It has been observed that the gradient noise $\nabla f_{\mathcal{S}_t}(\boldsymbol{x}_t) - \nabla f(\boldsymbol{x}_t)$ follows a normal distribution in CNN-based image classification models (Zhang et al., 2020; Kunstner et al., 2023). We confirmed experimentally that the search direction noise follows a normal distribution as well (see Section D.2). In addition, let $\boldsymbol{y}_t$ be the parameter updated by the gradient descent and $\boldsymbol{x}_{t+1}$ be the parameter updated by SHB at time $t$; i.e.,

$$\boldsymbol{y}_t := \boldsymbol{x}_t - \eta\nabla f(\boldsymbol{x}_t), \quad \boldsymbol{x}_{t+1} := \boldsymbol{x}_t - \eta\boldsymbol{m}_t = \boldsymbol{x}_t - \eta(\nabla f(\boldsymbol{x}_t) + \boldsymbol{\omega}_t^{\mathrm{SHB}}).$$

Then, according to Definition 2.1 and Assumption (A1), we have

$$\mathbb{E}_{\boldsymbol{\omega}_t^{\mathrm{SHB}}}[\boldsymbol{y}_{t+1}] = \mathbb{E}_{\boldsymbol{\omega}_t^{\mathrm{SHB}}}[\boldsymbol{y}_t] - \eta\nabla\mathbb{E}_{\boldsymbol{\omega}_t^{\mathrm{SHB}}}\left[f\left(\boldsymbol{y}_t - \eta\boldsymbol{\omega}_t^{\mathrm{SHB}}\right)\right] \tag{2}$$

$$= \mathbb{E}_{\boldsymbol{\omega}_t^{\mathrm{SHB}}}[\boldsymbol{y}_t] - \eta\nabla\mathbb{E}_{\boldsymbol{u}_t \sim \mathcal{N}\left(\boldsymbol{0}; \frac{1}{\sqrt{d}}I_d\right)}\left[f(\boldsymbol{y}_t - \psi^{\mathrm{SHB}}\boldsymbol{u}_t)\right]$$

$$= \mathbb{E}_{\boldsymbol{\omega}_t^{\mathrm{SHB}}}[\boldsymbol{y}_t] - \eta\nabla\hat{f}_{\eta\psi^{\mathrm{SHB}}}(\boldsymbol{y}_t).$$

(The derivation of Equation (2) is presented in Appendix C.3.) This shows that the function $\mathbb{E}_{\boldsymbol{\omega}_t^{\text{SHB}}}\left[f\left(\boldsymbol{y}_t - \eta\boldsymbol{\omega}_t^{\text{SHB}}\right)\right]$ is a smoothed version of $f$ with degree of smoothing $\eta\psi^{\text{SHB}}$. Furthermore, optimizing function $f$ with SHB is equivalent to optimizing function $\hat{f}_{\eta\psi^{\text{SHB}}}$ with gradient descent in the sense of expectation. Therefore, we can say that the degree of smoothing due to search direction noise in SHB is determined by $\delta^{\text{SHB}} = \eta\psi^{\text{SHB}}$, i.e., learning rate $\eta$, batch size $b$, momentum factor $\beta$, the variance of stochastic gradient $C_{\text{SHB}}^2$, and the upper bound of full gradient $K_{\text{SHB}}$ for SHB. The same argument holds for QHM. The degree of smoothing for each optimizer can be expressed as

$$\delta^{\text{SGD}} = \eta\sqrt{\frac{C_{\text{SGD}}^2}{b}}, \tag{3}$$

$$\delta^{\text{SHB}} = \eta\sqrt{\left(1 + \frac{\beta(\beta^2 - \beta + 1)}{(1 - \beta)^2}\right)\frac{C_{\text{SHB}}^2}{b} + \frac{\beta(\beta^2 - \beta + 1)}{(1 - \beta)^2}K_{\text{SHB}}^2}, \tag{4}$$

$$\delta^{\text{QHM}} = \eta\sqrt{\frac{1}{1 - \nu\beta}\frac{C_{\text{QHM}}^2}{b}}, \tag{5}$$

and if $\nu = 1$ in $\delta^{\text{QHM}}$, we obtain $\delta^{\text{NSHB}}$. Note that $\delta^{\text{SGD}}$ is the result derived by a previous study (Sato & Iiduka, 2023b). Since $\delta^{\text{SHB}}$ and $\delta^{\text{NSHB}}$ coincide with $\delta^{\text{SGD}}$ when $\beta = 0$ and $\nu = 0$, respectively, our results are an extension of their result. Since the terms $\frac{\beta(\beta^2 - \beta + 1)}{(1 - \beta^2)}$ and $\frac{1}{1 - \nu\beta}$ are monotone increasing for momentum factor $\beta$ or $\nu\beta$, we can say that a larger momentum factor leads to a greater degree of smoothing. In addition to the momentum factor, Equations (4) and (5) show that hyperparameters such as the learning rate and batch size also contribute to smoothing. Therefore, the learning rate, the batch size, and the momentum factor are interrelated, and they should be selected such that the degree of smoothing is appropriate. This finding is helpful in selecting appropriate hyperparameters. For example, from Lemma 2.1, if a large learning rate is used, a small momentum should obviously be used in order to obtain the appropriate degree of smoothing, i.e., one that is neither too large nor too small. Leclerc and Madry observed this phenomenon experimentally (Leclerc & Madry, 2020, Figure 4).

**Remark 3.1.** One may find it strange that, in equations (4) and (5), the upper bound $K_{\text{opt}}^2$ of the gradient norm appears only for $\delta^{\text{SHB}}$ and that it may be loose. In fact, the term $K_{\text{SHB}}^2$ plays an important role in our argument in Section 5, so we would like to add that this result is not arbitrary. As seen from the $\delta^{\text{QHM}}$ derivation, when expanding $\|\boldsymbol{\omega}_t^{\text{QHM}}\|^2$, we do not add unnecessary terms to the upper bound thanks to the convex combination property of the NSHB algorithm (see (20) in Appendix C.2 and Proposition A.1). This is one of our key technical contributions. In fact, by simply following the derivation of $\delta^{\text{SHB}}$, one can derive $\delta^{\text{QHM}}$ as follows:

$$\delta^{\text{QHM}} = \eta\sqrt{\left(1 + 4\nu^2\beta^2\right)\frac{C_{\text{QHM}}^2}{b} + 4\nu^2\beta^2 K_{\text{QHM}}^2}.$$

Thus, the reason QHM does not need an upper bound on the gradient to suppress $\delta^{\text{QHM}}$, even though QHM (NSHB) has a momentum term like SHB, is the presence of a convex combination in the QHM algorithm (see Algorithm 2). Furthermore, $K_{\text{SHB}}^2$ appears in the upper bound of SHB because the expansion of $\|\boldsymbol{\omega}_t^{\text{SHB}}\|^2$ cannot take advantage of the theoretically tractable properties of a convex combination. Because we experimentally demonstrated that SHB and QHM (NSHB) are completely different (see Figures 2 and Figure 4), we do not believe that this problem is simply due to a lack of good theoretical capture. Rather, we have shown for the first time, both theoretically and experimentally from the perspective of search direction noise, that the difference between the algorithms in terms of convex combination accounts for the difference in their respective performances. Of course, deriving $\delta^{\text{SHB}}$ without using $K_{\text{SHB}}^2$ should prove to be interesting and important future work.

Then, what are the magnitudes of the degree of smoothing $\delta^{\text{SGD}}$, $\delta^{\text{SHB}}$, and $\delta^{\text{QHM}}$, respectively? Since these include hyperparameters $\eta$, $b$, and $\beta$ as well as the unknowns $C_{\text{opt}}^2$ and $K_{\text{SHB}}^2$, it is necessary to estimate them in order to reveal the magnitude of the degree of smoothing. To estimate them, we provide some results in Section 4.

## 4 THEORETICAL ANALYSIS OF SHB AND QHM

We first use convergence analysis of SHB and QHM to clarify the relationship between batch size and the number of steps required for training. We then provide an equation for estimating the critical batch size and the variance of the stochastic gradient for an optimizer. To analyze SHB and QHM, we further assume that,

**Assumption 4.1.** *For all $\boldsymbol{x} \in \mathbb{R}^d$, there exists a positive real number $D(\boldsymbol{x})$ such that, for all $t \in \mathbb{N}$,*

$$\|\boldsymbol{x}_t - \boldsymbol{x}\| \leq D(\boldsymbol{x}).$$

Assumption 4.1 has been used to provide upper bounds on the performance measures when analyzing both convex and nonconvex optimization of DNNs (Kingma & Ba, 2015; Reddi et al., 2018; Zhuang et al., 2020). An example satisfying this assumption 4.1 is the boundedness condition of $(\boldsymbol{x}_t)_{t \in \mathbb{N}}$; i.e., there exists $D_1 > 0$ such that, for all $t \in \mathbb{N}, \|\boldsymbol{x}_t\| \leq D_1$. Then, we have that, for all $\boldsymbol{x} \in \mathbb{R}^d$ and all $t \in \mathbb{N}, \|\boldsymbol{x}_t - \boldsymbol{x}\| \leq \|\boldsymbol{x}_t\| + \|\boldsymbol{x}\| \leq D_1 + \|\boldsymbol{x}\| =: D(\boldsymbol{x})$, which implies that Assumption 4.1 holds.

### 4.1 CONVERGENCE ANALYSIS OF SHB AND QHM

We present convergence analyses of Algorithms 1 and 2 (The proofs of Theorems 4.1 and 4.2 are in Appendix A.4 and A.6 respectively).

**Theorem 4.1.** *Suppose that Assumptions (A1)−(A4) and 4.1 hold and consider the sequence $(\boldsymbol{x}_t)_{t \in \mathbb{N}}$ generated by SHB. Then, for all $\boldsymbol{x} \in \mathbb{R}^d$ and all $T \geq 1$, the following holds:*

$$\frac{1}{T} \sum_{t=0}^{T-1} \mathbb{E}\left[\langle \boldsymbol{x}_t - \boldsymbol{x}, \nabla f(\boldsymbol{x}_t)\rangle\right] \leq \frac{\|\boldsymbol{x}_0 - \boldsymbol{x}\|^2}{2\eta T} + \frac{\beta D(\boldsymbol{x})}{1-\beta}\sqrt{\frac{C_{\text{SHB}}^2}{b} + K_{\text{SHB}}^2}$$

$$+ \frac{\eta\left(\beta^2 - \beta + 1\right)}{2\beta(1-\beta)^2}\left(\frac{C_{\text{SHB}}^2}{b} + K_{\text{SHB}}^2\right).$$

**Theorem 4.2.** *Suppose that Assumptions (A1)−(A4) and 4.1 hold and consider the sequence $(\boldsymbol{x}_t)_{t \in \mathbb{N}}$ generated by QHM. Then, for all $\boldsymbol{x} \in \mathbb{R}^d$ and all $T \geq 1$, the following holds:*

$$\frac{1}{T} \sum_{t=0}^{T-1} \mathbb{E}\left[\langle \boldsymbol{x}_t - \boldsymbol{x}, \nabla f(\boldsymbol{x}_t)\rangle\right] \leq \frac{\|\boldsymbol{x}_0 - \boldsymbol{x}\|^2}{2\eta(1-\nu\beta)T} + \frac{\nu\beta D(\boldsymbol{x})}{1-\nu\beta}\sqrt{\frac{C_{\text{QHM}}^2}{b} + K_{\text{QHM}}^2}$$

$$+ \frac{\eta}{2(1-\nu\beta)}\left(\frac{C_{\text{QHM}}^2}{b} + K_{\text{QHM}}^2\right).$$

Convergence analysis for NSHB is performed using Theorem 4.2 with $\nu = 1$.

**Remark 4.1.** To illustrate the validity of the evaluation metrics in Theorems 4.1 and 4.2, we include Proposition A.2 in Appendix A. It implies that, if the upper bound of the inner product $\langle \boldsymbol{x}_t - \boldsymbol{x}, \nabla f(\boldsymbol{x}_t)\rangle$ becomes small, $\boldsymbol{x}_t$ comes to approximate a local minimizer of $f$ and that, if the upper bound is negative, $\boldsymbol{x}_t$ is simply a local minimizer of $f$. Therefore, Theorems 4.1 and 4.2 can be used to evaluate the inner products of unknown positivity.

### 4.2 ESTIMATION OF CRITICAL BATCH SIZE

We first define the stochastic first-order oracle (SFO) complexity, which is the stochastic gradient computation cost. If an optimizer uses batch size $b$ for training a DNNs, the optimizer computes $b$ stochastic gradients per step. If $T$ is the number of steps needed to train the DNNs, the optimizer has a stochastic gradient computation cost of $Tb$, which is the SFO complexity. We would like to minimize SFO complexity in order to minimize the computational cost. Previous studies (Shallue et al., 2019; Ma et al., 2018; McCandlish et al., 2018) have shown experimentally that the number of steps $T$ required to train a DNNs is halved when batch size $b$ is doubled, but this phenomenon is not observed beyond critical batch size $b^\star$. Therefore, the critical batch size is defined as the batch size that minimizes the SFO complexity for training, which is why it is desirable for the optimizer

to use the critical batch size that is the global minimizer of the SFO complexity $Tb$. Zhang et al. suggested that the critical batch size depends on the optimizer (Zhang et al., 2019), and Iiduka and Sato theoretically proved its existence and provided a formula for estimating its lower bound from the hyperparameters (Iiduka, 2022b; Sato & Iiduka, 2023a).

Letting $\epsilon > 0$ and using Theorems A.1, 4.1, and 4.2, we take $T_{\mathrm{opt}}$ satisfying $\frac{1}{T_{\mathrm{opt}}} \sum_{t=0}^{T_{\mathrm{opt}}-1} \mathbb{E}\left[\langle \boldsymbol{x}_t - \boldsymbol{x}, \nabla f(\boldsymbol{x}_t)\rangle\right] \leq \epsilon^2$ to be the number of steps required for training each optimizer. Thus, $\epsilon^2$ is a threshold and a stopping condition for training. Critical batch size $b_{\mathrm{opt}}^\star$ is defined as $b_{\mathrm{opt}}^\star := \mathrm{argmin}_{b\in[n]}T_{\mathrm{opt}}b$. From Theorems A.1, 4.1, and 4.2, we can derive the following proposition, which gives a lower bound on critical batch size $b_{\mathrm{opt}}^\star$. The proof of Proposition 4.1 and a more detailed discussion of its derivation are given in Appendix B.

**Proposition 4.1.** *Suppose that Assumptions (A1)−(A4) and 4.1 hold and consider SGD, SHB, and QHM. Let $\epsilon > 0$. Then, the following hold:*

$$b_{\mathrm{SGD}}^\star > \frac{\eta C_{\mathrm{SGD}}^2}{\epsilon^2}, \quad b_{\mathrm{SHB}}^\star > \frac{\eta(\beta^2 - \beta + 1)C_{\mathrm{SHB}}^2}{\beta(1-\beta)^2\epsilon^2}, \quad b_{\mathrm{QHM}}^\star > \frac{\eta C_{\mathrm{QHM}}^2}{(1-\nu\beta)\epsilon^2}.$$

Proposition 4.1 implies that the lower bound on the critical batch size of SHB is determined by learning rate $\eta$, the variance of the stochastic gradient $C_{\mathrm{SHB}}^2$, momentum factor $\beta$, and threshold $\epsilon$. It has been shown experimentally that there is a relationship between critical batch size and $\epsilon$, with more severe conditions increasing the critical batch size; see, for example, (Zhang et al., 2019). Our Proposition 4.1 theoretically supports their experimental results. It also provides a formula for estimating the lower bound for the critical batch size. In practice, however, estimating the critical batch size completely in advance is impossible because it involves an unknown, $C_{\mathrm{opt}}^2$. Nevertheless, this is an important proposition because it connects theory and experiment, and we can use it to back-calculate the variance of stochastic gradient $C_{\mathrm{opt}}^2$ (see Section 4.3).

### 4.3 ESTIMATION OF VARIANCE OF STOCHASTIC GRADIENT

We experimentally demonstrated the existence of a critical batch size. For different batch sizes, we measured the number of steps $T_{\mathrm{opt}}$ required for the gradient norm of the preceding $t$ steps at time $t$ to average less than $\epsilon = 0.5$ in training ResNet18 (He et al., 2016) on the CIFAR100 dataset (Krizhevsky, 2009). See Appendix B.4 for more details on the experiments discussed in this section and similar results on several datasets and models (see also Table 1).

A learning rate $\eta$ of 0.1 was used for all optimizers, with a momentum factor $\beta$ of 0.9 for SHB and NSHB. Figure 2 plots SFO complexity $T_{\mathrm{opt}}b$ versus $b$. The estimated critical batch sizes for SGD, SHB, and NSHB were $2^9$, $2^{10}$, and $2^9$, respectively. From Proposition 4.1 and these experimental results, we can estimate the upper bound on the variance of the stochastic gradient. For example, the variance of the stochastic gradient of SGD for training ResNet18 on the CIFAR100 dataset can be obtained as

$$C_{\mathrm{SGD}}^2 < \frac{b_{\mathrm{SGD}}^\star \epsilon^2}{\eta} = \frac{2^9 \cdot (0.5)^2}{0.1} = 1280.$$

Similar calculations for SHB and NSHB lead to $C_{\mathrm{SHB}}^2 < 25.3$ and $C_{\mathrm{NSHB}}^2 < 128$ (see Appendix B.5). Thus, adding a momentum term reduces the variance of the stochastic gradient, and the effect is seen especially in SHB for training ResNet18 on CIFAR100 dataset. We performed similar experiments for training WideResNet-28-10 (Zagoruyko & Komodakis, 2016) and MobileNetV2 (Sandler et al., 2018) on CIFAR100 dataset and training ResNet18 on CIFAR10 dataset (Krizhevsky, 2009). We also estimated an upper bound on the variance of the stochastic gradient $C_{\mathrm{opt}}^2$ from a similar discussion. The results are summarized in Table 1. We also experimentally observed an

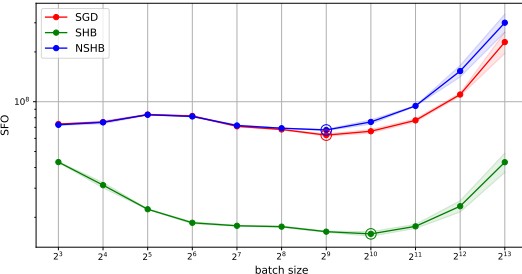

Figure 2: SFO complexities for SGD, SHB, and NSHB needed to train ResNet18 on CIFAR100 dataset versus batch size. The double circle symbols denote the critical batch sizes that minimize SFO complexity. The solid lines represent the mean value, and the shaded areas represent the maximum and minimum over three runs.

upper bound on the gradient norm (see Assumption (A4)) for training ResNet18 on the CIFAR100 dataset: $K_{\text{SGD}} = 4.528$, $K_{\text{SHB}} = 1.77$, and $K_{\text{NSHB}} = 4.5$ (see Appendix B.5). These values are used in our discussion of the smoothing property of SGD with momentum in Section 3.

Table 1: The variance of stochastic gradient $C_{\text{opt}}^2$ for training ResNet18, WideResNet-28-10, and MobileNetV2 on CIFAR100 and CIFAR10 datasets.

|  | CIFAR100 | | | CIFAR10 |
|---|---|---|---|---|
|  | ResNet18 | WideResNet-28-10 | MobileNetV2 | ResNet18 |
| $C_{\text{SGD}}^2$ | 1280 | 10 | 20 | 20 |
| $C_{\text{SHB}}^2$ | 25.3 | 0.79 | 6.33 | 0.79 |
| $C_{\text{NSHB}}^2$ | 128 | 1 | 2 | 2 |

## 5  DEGREE OF SMOOTHING AND GENERALIZABILITY

We have now completed estimating the unknowns in the equation for the degree of smoothing for each optimizer (3)-(5) we derived in Section 3. Using the value of the variance of the stochastic gradient and the upper bound of the gradient norm obtained in Section 4.3, we can obtain the degree of smoothing for each batch size. Figure 3 plots the degree of smoothing defined in (3)-(5) when $\eta = 0.1$ and $\beta = 0.9$ versus batch size.

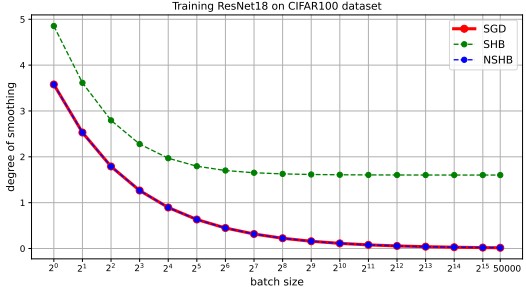 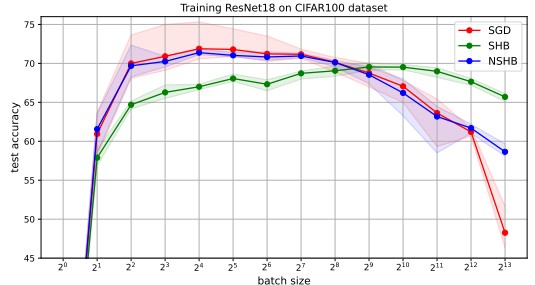

Figure 3: Degree of smoothing $\delta^{\text{SGD}}$, $\delta^{\text{SHB}}$, and $\delta^{\text{NSHB}}$ when $\eta = 0.1$ and $\beta = 0.9$ for SGD, SHB, and NSHB versus batch size in training ResNet18 on CIFAR100 dataset. See Appendix C.4 for details on calculating the degree of smoothing and a logarithmic graph version (see also Figure 1).

Figure 4: Test accuracy for SGD, SHB, and NSHB versus batch size in training ResNet18 on CIFAR100 dataset. The solid line represents the mean value, and the shaded area represents the maximum and minimum over seven runs.

**Why and when momentum improves generalizability.** We measured the test accuracy for 11 batch sizes for 200 epochs for training ResNet18 for SGD, SHB, and NSHB on the CIFAR100 dataset. As shown in Figure 4, the generalizability of SGD and NSHB deteriorated as the batch size was increased, whereas that of SHB remained stable. If the degree of smoothing is not sufficient, the optimizer will fall into a sharp local optimal solution, and generalizability will be compromised. Therefore, the reason SHB outperforms SGD and NSHB when the batch is large is that the degree of smoothing of SGD and NSHB approaches zero, whereas SHB has a reasonably large degree of smoothing even for large batches. This is also why the gap in generalizability between SGD and SHB is more pronounced for large batches as observed in several previous studies (Shallue et al., 2019; Jelassi & Li, 2022; Kunstner et al., 2023).

Figure 4 also shows that SHB has stable generalizability for all batch sizes, but accuracy never exceeds 70%, which is highest accuracy of SGD and NSHB. This can also be explained by the greater or lesser degree of smoothing shown in Figure 3: SHB does not decrease in degree of smoothing with increasing batch size but always has a greater degree of smoothing than SGD and NSHB. From Lemma 2.1, a too large degree of smoothing leads to too large deviations from the original function. Therefore, the reason that the test accuracy of SHB never exceeds 70% is that the degree of smoothing for SHB is always slightly greater than the appropriate value. Then, we can say that the degree of smoothing from $b = 2^3$ to $b = 2^8$, where SGD and NSHB achieve high test accuracies, is an appropriate value for training on the ResNet18 on CIFAR100 dataset.

In summary, momentum improves generalizability when the batch size is large, but deteriorates generalizability when the batch size is small. There is an impressive correlation between the degree of smoothing and model generalizability, which is why we can say that the degree of smoothing introduced by the optimizer's search direction noise dominates model training and generalizability.

**Remark 5.1.** Recently, Wang et al. showed that when the learning rate is small, there is no significant difference in generalization performance between SGD and SGD with momentum (Wang et al., 2024). SGD with momentum in their paper is NSHB in our paper (see (Wang et al., 2024, Definition 2.3) and our Algorithm 2). Our results show that SGD and NSHB have same degrees of smoothing (see Figure 3), which results in nearly same test accuracy (see Figure 4). Therefore, our results do not conflict with theirs.

**Remark 5.2.** Let us explain the relationship between the degree of smoothing and expected loss. Previous studies (Keskar et al., 2017; Izmailov et al., 2018; Li et al., 2018) have shown that the sharpness around the approximate solution to which the optimizer converges is closely related to the generalization performance of the model, i.e., the expected loss. Sato and Iiduka (Sato & Iiduka, 2023b) experimentally demonstrated that sharpness and the degree of smoothing introduced by search direction noise are inextricably linked. That is, when the degree of smoothing is small (resp. large), sharpness is large (resp. small). Thus, the degree of smoothing due to empirical loss is related to expected loss, a measure of true generalization performance, via sharpness. In particular, the degree of smoothing is correlated with generalization performance, as shown by a previous study (Sato & Iiduka, 2023b) and our experimental results.

**Resolving the contradiction between momentum and stochastic noise.** Figure 3 shows that SHB always has a greater degree of smoothing than SGD. Therefore, for SHB, which is often used experimentally, adding momentum increases the search direction noise. We can thus say that adding momentum reduces the variance of the stochastic gradient (see Section 4.3) and conversely increases the degree of smoothing introduced by the search direction noise. This is why the arguments that "adding momentum should reduce stochastic noise" and that "stochastic noise leads to generalizability" do not conflict, which resolves the contradiction. Figure 3 also shows that NSHB has the same degree of smoothing as SGD. Thus, for NSHB, which is rarely used experimentally, adding momentum does not contribute to an increase in the degree of smoothing. In fact, the performances of SGD and NSHB are very similar (see Figures 2 and 4). This not only demonstrates that the degree of smoothing is a hidden factor governing the training of the model but also that the reason NSHB is not as good and not used experimentally as often as SHB is that the degree of smoothing does not differ from that of SGD despite the addition of momentum. Thus, for NSHB, there was no contradiction regarding the momentum term and stochastic noise.

# 6 CONCLUSION

Our investigation of the smoothing properties of SGD with momentum resolved the contradiction between momentum and stochastic noise, namely that adding momentum reduces gradient noise and conversely increases search direction noise, which contributes to smoothing of the objective function. It also showed that the degree of smoothing is determined by the hyperparameters such as the momentum factor. Through convergence analysis and discussion of critical batch size estimation, we derived the degree of smoothing numerically and found an impressive correlation between the degree of smoothing and model generalizability. Specifically, too large or too small a degree of smoothing leads to poor generalizability, whereas a moderate one leads to high generalizability. From this perspective, we showed that SHB and NSHB are completely different, that NSHB has almost no experimental value, and that the momentum factor in SHB maintains a high degree of smoothing even when the batch is large. The relationship between the degree of smoothing and model generalizability is, so to speak, a hidden factor in DNNs training, and it helps in selecting the optimal hyperparameters and understanding the training dynamics of a DNNs. Finally, we emphasize that the degree of smoothing introduced by search direction noise is determined by several hyperparameters, including the learning rate and batch size, that are easier to grasp than sharpness. They are thus useful as a new measure of generalization performance. Deriving or estimating the optimal degree of smoothing for generalization performance is important future work, which, if accomplished, will reduce the huge computational cost of hyperparameter tuning.

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

# A  CONVERGENCE ANALYSIS OF STOCHASTIC GRADIENT DESCENT (SGD), STOCHASTIC HEAVY BALL (SHB), AND QUASIHYPERBOLIC MOMENTUM (QHM)

## A.1  VERIFICATION OF THE VALIDITY OF ASSUMPTION (A2)(II)

We measured the value $\|\mathsf{G}_{\boldsymbol{\xi}_t}(\boldsymbol{x}_t) - \nabla f(\boldsymbol{x}_t)\|$ 500 times using ResNet18 trained on the CIFAR100 dataset by SGD, SHB, and NSHB (10,000 steps) to determine its magnitude. As seen in Figure 5, the variance of stochastic gradient $C_{\mathrm{opt}}^2$ depends on the optimizer. In fact, the values for SHB training are smaller than the ones for SGD training.

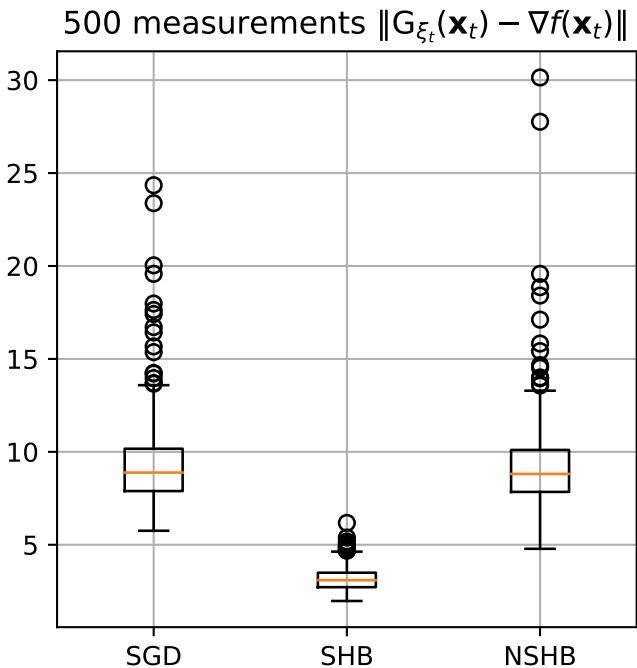

Figure 5: Box plot of 500 measurements of $\|\mathsf{G}_{\boldsymbol{\xi}_t}(\boldsymbol{x}_t) - \nabla f(\boldsymbol{x}_t)\|$ using ResNet18 trained on CIFAR100 dataset by SGD, SHB, and NSHB (10,000 steps). The code used is available at our anonymous GitHub repository (https://anonymous.4open.science/r/role-of-momentum).

Note that although these experimental results are important in motivating Assumption (A2)(ii), $C_{\mathrm{opt}}^2$ cannot be estimated from these results alone since $C_{\mathrm{opt}}^2$ is a constant satisfying $\|\mathsf{G}_{\boldsymbol{\xi}_t}(\boldsymbol{x}_t) - \nabla f(\boldsymbol{x}_t)\| \leq C_{\mathrm{opt}}^2$ for any $t \in \mathbb{N}$. See Section 4 for a discussion of our estimation of $C_{\mathrm{opt}}^2$.

## A.2  PROPOSITIONS AND LEMMAS FOR ANALYSES

**Proposition A.1.** *For all $\boldsymbol{x}, \boldsymbol{y} \in \mathbb{R}^d$ and all $\alpha \in \mathbb{R}$, the following holds:*

$$\|\alpha \boldsymbol{x} + (1-\alpha)\boldsymbol{y}\|^2 = \alpha\|\boldsymbol{x}\|^2 + (1-\alpha)\|\boldsymbol{y}\|^2 - \alpha(1-\alpha)\|\boldsymbol{x} - \boldsymbol{y}\|^2.$$

*Proof.* Since $2\langle \boldsymbol{x}, \boldsymbol{y} \rangle = \|\boldsymbol{x}\|^2 + \|\boldsymbol{y}\|^2 - \|\boldsymbol{x} - \boldsymbol{y}\|^2$ holds, for all $\boldsymbol{x}, \boldsymbol{y} \in \mathbb{R}^d$ and all $\alpha \in \mathbb{R}$,

$$
\begin{aligned}
\|\alpha \boldsymbol{x} + (1-\alpha)\boldsymbol{y}\|^2 &= \alpha\|\boldsymbol{x}\|^2 + 2\alpha(1-\alpha)\langle \boldsymbol{x}, \boldsymbol{y} \rangle + (1-\alpha)^2\|\boldsymbol{y}\|^2 \\
&= \alpha\|\boldsymbol{x}\|^2 + \alpha(1-\alpha)(\|\boldsymbol{x}\|^2 + \|\boldsymbol{y}\|^2 - \|\boldsymbol{x} - \boldsymbol{y}\|^2) + (1-\alpha)^2\|\boldsymbol{y}\|^2 \\
&= \alpha\|\boldsymbol{x}\|^2 + (1-\alpha)\|\boldsymbol{y}\|^2 - \alpha(1-\alpha)\|\boldsymbol{x} - \boldsymbol{y}\|^2.
\end{aligned}
$$

This completes the proof. □

The following proposition describes the relationship between the stationary point problem and variational inequality.

**Proposition A.2.** *Suppose that $f : \mathbb{R}^d \to \mathbb{R}$ is continuously differentiable and $\boldsymbol{x}^*$ is a stationary point of $f$. Then, $\nabla f(\boldsymbol{x}^*) = \boldsymbol{0}$ is equivalent to the following variational inequality: for all $\boldsymbol{x} \in \mathbb{R}^d$,*

$$\langle \nabla f(\boldsymbol{x}^*), \boldsymbol{x} - \boldsymbol{x}^* \rangle \geq 0.$$

*Proof.* Suppose that $\boldsymbol{x} \in \mathbb{R}^d$ satisfies $\nabla f(\boldsymbol{x}) = \boldsymbol{0}$. Then, for all $\boldsymbol{y} \in \mathbb{R}^d$,

$$\langle \nabla f(\boldsymbol{x}), \boldsymbol{y} - \boldsymbol{x} \rangle \geq 0.$$

Suppose that $\boldsymbol{x} \in \mathbb{R}^d$ satisfies $\langle \nabla f(\boldsymbol{x}), \boldsymbol{y} - \boldsymbol{x} \rangle \geq 0$ for all $\boldsymbol{y} \in \mathbb{R}^d$. Let $\boldsymbol{y} := \boldsymbol{x} - \nabla f(\boldsymbol{x})$. Then we have

$$0 \leq \langle \nabla f(\boldsymbol{x}), \boldsymbol{y} - \boldsymbol{x} \rangle = -\|\nabla f(\boldsymbol{x})\|^2.$$

Hence,

$$\nabla f(\boldsymbol{x}) = \boldsymbol{0}.$$

This completes the proof. $\qquad\square$

**Lemma A.1.** *Suppose that (A2)(ii) and (A3) hold for all $t \in \mathbb{N}$; then,*

$$\mathbb{E}_{\boldsymbol{\xi}_t} \left[ \|\nabla f_{\mathcal{S}_t}(\boldsymbol{x}_t) - \nabla f(\boldsymbol{x}_t)\|^2 \right] \leq \frac{C_{\mathrm{opt}}^2}{b}.$$

*Proof.* Let $t \in \mathbb{N}$ and $\boldsymbol{\xi}_t := (\xi_{t,1}, \cdots, \xi_{t,b})^\top$. Then, (A2)(ii) and (A3) guarantee that

$$
\mathbb{E}_{\boldsymbol{\xi}_t} \left[ \|\nabla f_{\mathcal{S}_t}(\boldsymbol{x}_t) - \nabla f(\boldsymbol{x}_t)\|^2 \big| \boldsymbol{x}_t \right] = \mathbb{E}_{\boldsymbol{\xi}_t} \left[ \left\| \frac{1}{b} \sum_{i=1}^b \mathsf{G}_{\xi_{t,i}}(\boldsymbol{x}_t) - \nabla f(\boldsymbol{x}_t) \right\|^2 \right]
$$

$$
= \mathbb{E}_{\boldsymbol{\xi}_t} \left[ \left\| \frac{1}{b} \sum_{i=1}^b \mathsf{G}_{\xi_{t,i}}(\boldsymbol{x}_t) - \frac{1}{b} \sum_{i=1}^b \nabla f(\boldsymbol{x}_t) \right\|^2 \right]
$$

$$
= \mathbb{E}_{\boldsymbol{\xi}_t} \left[ \left\| \frac{1}{b} \sum_{i=1}^b \left( \mathsf{G}_{\xi_{t,i}}(\boldsymbol{x}_t) - \nabla f(\boldsymbol{x}_t) \right) \right\|^2 \right]
$$

$$
= \frac{1}{b^2} \mathbb{E}_{\boldsymbol{\xi}_t} \left[ \left\| \sum_{i=1}^b \left( \mathsf{G}_{\xi_{t,i}}(\boldsymbol{x}_t) - \nabla f(\boldsymbol{x}_t) \right) \right\|^2 \right]
$$

$$
= \frac{1}{b^2} \mathbb{E}_{\boldsymbol{\xi}_t} \left[ \sum_{i=1}^b \left\| \mathsf{G}_{\xi_{t,i}}(\boldsymbol{x}_t) - \nabla f(\boldsymbol{x}_t) \right\|^2 \right]
$$

$$
\leq \frac{C_{\mathrm{opt}}^2}{b}.
$$

This completes the proof. $\qquad\square$

**Lemma A.2.** *Suppose that Assumptions (A2) and (A4) hold, then for all $t \in \mathbb{N}$,*

$$\mathbb{E} \left[ \|\nabla f_{\mathcal{S}_t}(\boldsymbol{x}_t)\|^2 \right] \leq \frac{C_{\mathrm{opt}}^2}{b} + K_{\mathrm{opt}}^2,$$

*where $\mathbb{E} = \mathbb{E}_{\boldsymbol{\xi}_0} \mathbb{E}_{\boldsymbol{\xi}_1} \cdots \mathbb{E}_{\boldsymbol{\xi}_t}$.*

*Proof.* Let $t \in \mathbb{N}$. From (A2)(i), we obtain

$$
\mathbb{E}_{\boldsymbol{\xi}_t} \left[ \|\nabla f_{\mathcal{S}_t}(\boldsymbol{x}_t)\|^2 \big| \boldsymbol{x}_t \right] = \mathbb{E}_{\boldsymbol{\xi}_t} \left[ \|\nabla f_{\mathcal{S}_t}(\boldsymbol{x}_t) - \nabla f(\boldsymbol{x}_t) + \nabla f(\boldsymbol{x}_t)\|^2 \big| \boldsymbol{x}_t \right]
$$

$$
= \mathbb{E}_{\boldsymbol{\xi}_t} \left[ \|\nabla f_{\mathcal{S}_t}(\boldsymbol{x}_t) - \nabla f(\boldsymbol{x}_t)\|^2 \big| \boldsymbol{x}_t \right] + \mathbb{E} \left[ \|\nabla f(\boldsymbol{x}_t)\|^2 \big| \boldsymbol{x}_t \right]
$$

$$
+ 2 \mathbb{E}_{\boldsymbol{\xi}_t} \left[ \langle \nabla f_{\mathcal{S}_t}(\boldsymbol{x}_t) - \nabla f(\boldsymbol{x}_t), \nabla f(\boldsymbol{x}_t) \rangle \big| \boldsymbol{x}_t \right]
$$

$$
= \mathbb{E} \left[ \|\nabla f_{\mathcal{S}_t}(\boldsymbol{x}_t) - \nabla f(\boldsymbol{x}_t)\|^2 \big| \boldsymbol{x}_t \right] + \|\nabla f(\boldsymbol{x}_t)\|^2,
$$

which, together with (A2)(ii), (A4), Lemma A.1, and $\mathbb{E} = \mathbb{E}_{\boldsymbol{\xi}_0}\mathbb{E}_{\boldsymbol{\xi}_1}\cdots\mathbb{E}_{\boldsymbol{\xi}_t}$ implies that

$$\mathbb{E}\left[\|\nabla f_{\mathcal{S}_t}(\boldsymbol{x}_t)\|^2\right] \le \frac{C_{\text{opt}}^2}{b} + K_{\text{opt}}^2.$$

This completes the proof. □

### A.3 LEMMAS FOR THE CONVERGENCE ANALYSIS OF SHB

**Lemma A.3.** *Suppose that Assumptions (A2)(ii), (A3), and (A4) hold, then for all $t \in \mathbb{N}$,*

$$\mathbb{E}\left[\|\boldsymbol{m}_t\|\right] \le \frac{1}{1-\beta}\sqrt{\frac{C_{\text{SHB}}^2}{b} + K_{\text{SHB}}^2}.$$

*Proof.* Let $(\boldsymbol{x}_t)_{t\in\mathbb{N}}$ be the sequence generated by SHB and $t \in \mathbb{N}$. The definition of $\boldsymbol{m}_t$ implies that

$$\begin{aligned}
\boldsymbol{m}_t &:= \nabla f_{S_t}(\boldsymbol{x}_t) + \beta\boldsymbol{m}_{t-1} \\
&= \nabla f_{S_t}(\boldsymbol{x}_t) + \beta(\nabla f_{S_{t-1}}(\boldsymbol{x}_{t-1}) + \beta\boldsymbol{m}_{t-2}) \\
&\;\;\vdots \\
&= \nabla f_{S_t}(\boldsymbol{x}_t) + \beta\nabla f_{S_{t-1}}(\boldsymbol{x}_{t-1}) + \beta^2\nabla f_{S_{t-2}}(\boldsymbol{x}_{t-2}) + \cdots + \beta^t\nabla f_{S_0}(\boldsymbol{x}_0).
\end{aligned}$$

By using the triangle inequality, we obtain

$$\begin{aligned}
\|\boldsymbol{m}_t\| &= \|\nabla f_{S_t}(\boldsymbol{x}_t) + \beta\nabla f_{S_{t-1}}(\boldsymbol{x}_{t-1}) + \beta^2\nabla f_{S_{t-2}}(\boldsymbol{x}_{t-2}) + \cdots + \beta^t\nabla f_{S_0}(\boldsymbol{x}_0)\| \\
&\le \|\nabla f_{S_t}(\boldsymbol{x}_t)\| + \beta\|\nabla f_{S_{t-1}}(\boldsymbol{x}_{t-1})\| + \beta^2\|\nabla f_{S_{t-2}}(\boldsymbol{x}_{t-2})\| + \cdots + \beta^t\|\nabla f_{S_0}(\boldsymbol{x}_0)\|.
\end{aligned}$$

From Lemma A.2,

$$\begin{aligned}
\mathbb{E}\left[\|\boldsymbol{m}_t\|\right] &\le \sqrt{\frac{C_{\text{SHB}}^2}{b} + K_{\text{SHB}}^2} + \beta\sqrt{\frac{C_{\text{SHB}}^2}{b} + K_{\text{SHB}}^2} + \beta^2\sqrt{\frac{C_{\text{SHB}}^2}{b} + K_{\text{SHB}}^2} + \cdots \beta^t\sqrt{\frac{C_{\text{SHB}}^2}{b} + K_{\text{SHB}}^2} \\
&= \frac{(1-\beta^t)}{1-\beta}\sqrt{\frac{C_{\text{SHB}}^2}{b} + K_{\text{SHB}}^2} \\
&\le \frac{1}{1-\beta}\sqrt{\frac{C_{\text{SHB}}^2}{b} + K_{\text{SHB}}^2}.
\end{aligned}$$

This completes the proof. □

**Lemma A.4.** *Suppose that Assumptions (A2) and (A4) hold, then for all $t \in \mathbb{N}$,*

$$\mathbb{E}\left[\|\boldsymbol{m}_t\|^2\right] \le \frac{\beta^2 - \beta + 1}{\beta(1-\beta)^2}\left(\frac{C_{\text{SHB}}^2}{b} + K_{\text{SHB}}^2\right).$$

*Proof.* Let $(\boldsymbol{x}_t)_{t\in\mathbb{N}}$ be the sequence generated by SHB and $t \in \mathbb{N}$. Proposition A.1 guarantees that

$$\begin{aligned}
&\beta(1-\beta)\|\nabla f_{S_t}(\boldsymbol{x}_t) + \boldsymbol{m}_{t-1}\|^2 \\
&= \beta\|\nabla f_{S_t}(\boldsymbol{x}_{t-1})\|^2 + (1-\beta)\|\boldsymbol{m}_{t-1}\|^2 - \|\beta\nabla f_{S_t}(\boldsymbol{x}_t) - (1-\beta)\boldsymbol{m}_{t-1}\|^2 \\
&\le \beta\|\nabla f_{S_t}(\boldsymbol{x}_t)\|^2 + (1-\beta)\|\boldsymbol{m}_{t-1}\|^2.
\end{aligned}$$

Hence,

$$\|\nabla f_{S_t}(\boldsymbol{x}_t) + \boldsymbol{m}_{t-1}\|^2 \le \frac{1}{1-\beta}\|\nabla f_{S_t}(\boldsymbol{x}_t)\|^2 + \frac{1}{\beta}\|\boldsymbol{m}_{t-1}\|^2. \tag{6}$$

On the other hand,

$$\|\nabla f_{S_t}(\boldsymbol{x}_t) + \boldsymbol{m}_{t-1}\|^2 = \|\nabla f_{S_t}(\boldsymbol{x}_t)\|^2 + 2\langle\nabla f_{S_t}(\boldsymbol{x}_t), \boldsymbol{m}_{t-1}\rangle + \|\boldsymbol{m}_{t-1}\|^2. \tag{7}$$

From (6) and (7), we obtain

$$\|\nabla f_{S_t}(\boldsymbol{x}_t)\|^2 + 2\langle\nabla f_{S_t}(\boldsymbol{x}_t), \boldsymbol{m}_{t-1}\rangle + \|\boldsymbol{m}_{t-1}\|^2 \le \frac{1}{1-\beta}\|\nabla f_{S_t}(\boldsymbol{x}_t)\|^2 + \frac{1}{\beta}\|\boldsymbol{m}_{t-1}\|^2.$$

Therefore,

$$2\langle \nabla f_{S_t}(\boldsymbol{x}_t), \boldsymbol{m}_{t-1}\rangle \leq \frac{\beta}{1-\beta}\|\nabla f_{S_t}(\boldsymbol{x}_t)\|^2 + \frac{1-\beta}{\beta}\|\boldsymbol{m}_{t-1}\|^2. \tag{8}$$

The definition of $\boldsymbol{m}_t$ implies that

$$\begin{aligned}
\|\boldsymbol{m}_t\|^2 &= \|\nabla f_{S_t}(\boldsymbol{x}_t) + \beta \boldsymbol{m}_{t-1}\|^2 \\
&= \|\nabla f_{S_t}(\boldsymbol{x}_t)\|^2 + 2\beta\langle \nabla f_{S_t}(\boldsymbol{x}_t), \boldsymbol{m}_{t-1}\rangle + \beta^2\|\boldsymbol{m}_{t-1}\|^2.
\end{aligned} \tag{9}$$

From (8) and (9), we obtain

$$\begin{aligned}
\|\boldsymbol{m}_t\|^2 &\leq \frac{\beta^2 - \beta + 1}{1-\beta}\|\nabla f_{S_t}(\boldsymbol{x}_t)\|^2 + \left(\beta^2 - \beta + 1\right)\|\boldsymbol{m}_{t-1}\|^2 \\
&\leq \frac{\beta^2 - \beta + 1}{1-\beta}\|\nabla f_{S_t}(\boldsymbol{x}_t)\|^2 \\
&\quad + \left(\beta^2 - \beta + 1\right)\left\{\frac{\beta^2 - \beta + 1}{1-\beta}\|\nabla f_{S_{t-1}}(\boldsymbol{x}_{t-1})\|^2 + \left(\beta^2 - \beta + 1\right)\|\boldsymbol{m}_{t-2}\|^2\right\} \\
&\leq \frac{\beta^2 - \beta + 1}{1-\beta}\|\nabla f_{S_t}(\boldsymbol{x}_t)\|^2 + \cdots + \frac{\beta^2 - \beta + 1}{1-\beta}\left(\beta^2 - \beta + 1\right)^t\|\nabla f_{S_0}(\boldsymbol{x}_0)\|^2.
\end{aligned}$$

By taking the total expectation on both sides, from Lemma A.2, we obtain

$$\begin{aligned}
\mathbb{E}\left[\|\boldsymbol{m}_t\|^2\right] &\leq \frac{\beta^2 - \beta + 1}{1-\beta}\mathbb{E}\left[\|\nabla f_{S_t}(\boldsymbol{x}_t)\|^2\right] + \cdots + \frac{\beta^2 - \beta + 1}{1-\beta}\left(\beta^2 - \beta + 1\right)^t\mathbb{E}\left[\|\nabla f_{S_0}(\boldsymbol{x}_0)\|^2\right] \\
&\leq \frac{\beta^2 - \beta + 1}{1-\beta}\left(\frac{C_{\text{SHB}}^2}{b} + K_{\text{SHB}}^2\right) + \cdots + \frac{\beta^2 - \beta + 1}{1-\beta}\left(\beta^2 - \beta + 1\right)^t\left(\frac{C_{\text{SHB}}^2}{b} + K_{\text{SHB}}^2\right) \\
&= \frac{\beta^2 - \beta + 1}{1-\beta}\left(\frac{C_{\text{SHB}}^2}{b} + K_{\text{SHB}}^2\right)\cdot\frac{1 - \left(\beta^2 - \beta + 1\right)^{t+1}}{1 - \left(\beta^2 - \beta + 1\right)} \\
&\leq \frac{\beta^2 - \beta + 1}{1-\beta}\left(\frac{C_{\text{SHB}}^2}{b} + K_{\text{SHB}}^2\right)\cdot\frac{1}{1 - \left(\beta^2 - \beta + 1\right)} \\
&= \frac{\beta^2 - \beta + 1}{1-\beta}\left(\frac{C_{\text{SHB}}^2}{b} + K_{\text{SHB}}^2\right)\cdot\frac{1}{\beta(1-\beta)} \\
&= \frac{\beta^2 - \beta + 1}{\beta(1-\beta)^2}\left(\frac{C_{\text{SHB}}^2}{b} + K_{\text{SHB}}^2\right).
\end{aligned}$$

This completes the proof. $\qquad\square$

### A.4 PROOF OF THEOREM 4.1

*Proof.* Let $\boldsymbol{x} \in \mathbb{R}^d$ and $t \in \mathbb{N}$. The definition of $\boldsymbol{x}_{t+1}$ implies that

$$\begin{aligned}
\|\boldsymbol{x}_{t+1} - \boldsymbol{x}\|^2 &= \|(\boldsymbol{x}_t - \eta\boldsymbol{m}_t) - \boldsymbol{x}\|^2 \\
&= \|\boldsymbol{x}_t - \boldsymbol{x}\|^2 - 2\eta\langle \boldsymbol{x}_t - \boldsymbol{x}, \boldsymbol{m}_t\rangle + \eta^2\|\boldsymbol{m}_t\|^2 \\
&= \|\boldsymbol{x}_t - \boldsymbol{x}\|^2 - 2\eta\langle \boldsymbol{x}_t - \boldsymbol{x}, \nabla f_{S_t}(\boldsymbol{x}_t)\rangle + 2\eta\beta\langle \boldsymbol{x} - \boldsymbol{x}_t, \boldsymbol{m}_{t-1}\rangle + \eta^2\|\boldsymbol{m}_t\|^2.
\end{aligned}$$

We then have

$$\begin{aligned}
&\mathbb{E}_{\boldsymbol{\xi}_t}\left[\langle \boldsymbol{x}_t - \boldsymbol{x}, \nabla f_{S_t}(\boldsymbol{x}_t)\rangle\Big|\boldsymbol{x}_t\right] \\
&= \mathbb{E}_{\boldsymbol{\xi}_t}\left[\frac{1}{2\eta}\left(\|\boldsymbol{x}_t - \boldsymbol{x}\|^2 - \|\boldsymbol{x}_{t+1} - \boldsymbol{x}\|^2\right) + \beta\langle \boldsymbol{x} - \boldsymbol{x}_t, \boldsymbol{m}_{t-1}\rangle + \frac{\eta}{2}\|\boldsymbol{m}_t\|^2\Big|\boldsymbol{x}_t\right].
\end{aligned}$$

On the other hand, Assumptions (A2)(ii) and (A3) guarantees that

$$\begin{aligned}
\mathbb{E}_{\boldsymbol{x}_t}\left[\mathbb{E}_{\boldsymbol{\xi}_t}\left[\langle \boldsymbol{x}_t - \boldsymbol{x}, \nabla f_{S_t}(\boldsymbol{x}_t)\rangle\Big|\boldsymbol{x}_t\right]\right] &= \mathbb{E}_{\boldsymbol{x}_t}\left[\left\langle \boldsymbol{x}_t - \boldsymbol{x}, \mathbb{E}_{\boldsymbol{\xi}_t}\left[\nabla f_{S_t}(\boldsymbol{x}_t)\Big|\boldsymbol{x}_t\right]\right\rangle\right] \\
&= \mathbb{E}_{\boldsymbol{x}_t}\left[\langle \boldsymbol{x}_t - \boldsymbol{x}, \nabla f(\boldsymbol{x}_t)\rangle\right].
\end{aligned}$$

Hence, by taking the total expectation on both sides, we obtain

$$\mathbb{E}\left[\langle \boldsymbol{x}_t - \boldsymbol{x}, \nabla f(\boldsymbol{x}_t) \rangle\right]$$

$$= \frac{1}{2\eta} \left(\mathbb{E}\left[\|\boldsymbol{x}_t - \boldsymbol{x}\|^2\right] - \mathbb{E}\left[\|\boldsymbol{x}_{t+1} - \boldsymbol{x}\|^2\right]\right) + \beta\mathbb{E}\left[\langle \boldsymbol{x} - \boldsymbol{x}_t, \boldsymbol{m}_{t-1} \rangle\right] + \frac{\eta}{2}\mathbb{E}\left[\|\boldsymbol{m}_t\|^2\right].$$

According to Lemmas A.3 and A.4, Assumption 4.1, and the Cauchy-Schwarz inequality,

$$\mathbb{E}\left[\langle \boldsymbol{x}_t - \boldsymbol{x}, \nabla f(\boldsymbol{x}_t) \rangle\right] \leq \frac{1}{2\eta} \left(\mathbb{E}\left[\|\boldsymbol{x}_t - \boldsymbol{x}\|^2\right] - \mathbb{E}\left[\|\boldsymbol{x}_{t+1} - \boldsymbol{x}\|^2\right]\right)$$

$$+ \frac{\beta D(\boldsymbol{x})}{1 - \beta}\sqrt{\frac{C_{\mathrm{SHB}}^2}{b} + K_{\mathrm{SHB}}^2} + \frac{\eta\left(\beta^2 - \beta + 1\right)}{2\beta(1 - \beta)^2}\left(\frac{C_{\mathrm{SHB}}^2}{b} + K_{\mathrm{SHB}}^2\right).$$

Summing over $t$ from $t = 0$ to $t = T - 1$, we obtain

$$\sum_{t=0}^{T-1}\mathbb{E}\left[\langle \boldsymbol{x}_t - \boldsymbol{x}, \nabla f(\boldsymbol{x}_t) \rangle\right] \leq \frac{1}{2\eta}\left(\mathbb{E}\left[\|\boldsymbol{x}_0 - \boldsymbol{x}\|^2\right] - \mathbb{E}\left[\|\boldsymbol{x}_T - \boldsymbol{x}\|^2\right]\right)$$

$$+ \frac{\beta D(\boldsymbol{x})}{1 - \beta}\sqrt{\frac{C_{\mathrm{SHB}}^2}{b} + K_{\mathrm{SHB}}^2}\,T + \frac{\eta\left(\beta^2 - \beta + 1\right)}{2\beta(1 - \beta)^2}\left(\frac{C_{\mathrm{SHB}}^2}{b} + K_{\mathrm{SHB}}^2\right)T.$$

Therefore,

$$\frac{1}{T}\sum_{t=0}^{T-1}\mathbb{E}\left[\langle \boldsymbol{x}_t - \boldsymbol{x}, \nabla f(\boldsymbol{x}_t) \rangle\right]$$

$$\leq \frac{\|\boldsymbol{x}_0 - \boldsymbol{x}\|^2}{2\eta T} + \frac{\beta D(\boldsymbol{x})}{1 - \beta}\sqrt{\frac{C_{\mathrm{SHB}}^2}{b} + K_{\mathrm{SHB}}^2} + \frac{\eta\left(\beta^2 - \beta + 1\right)}{2\beta(1 - \beta)^2}\left(\frac{C_{\mathrm{SHB}}^2}{b} + K_{\mathrm{SHB}}^2\right).$$

This completes the proof. $\qquad\square$

### A.5 LEMMA FOR CONVERGENCE ANALYSIS OF QHM

**Lemma A.5.** *Suppose that Assumptions (A2) and (A4) hold, then for all $t \in \mathbb{N}$,*

$$\mathbb{E}\left[\|\boldsymbol{d}_t\|^2\right] \leq \frac{C_{\mathrm{QHM}}^2}{b} + K_{\mathrm{QHM}}^2.$$

*Proof.* The convexity of $\|\cdot\|^2$, together with the definition of $\boldsymbol{d}_t$ and Lemma A.2, guarantees that, for all $t \in \mathbb{N}$,

$$\mathbb{E}\left[\|\boldsymbol{d}_t\|^2\right] \leq \nu\beta\mathbb{E}\left[\|\boldsymbol{d}_{t-1}\|^2\right] + (1 - \nu\beta)\mathbb{E}\left[\|\nabla f_{\mathcal{S}_t}(\boldsymbol{x}_t)\|^2\right]$$

$$\leq \nu\beta\mathbb{E}\left[\|\boldsymbol{d}_{t-1}\|^2\right] + (1 - \nu\beta)\left(\frac{C_{\mathrm{QHM}}^2}{b} + K_{\mathrm{QHM}}^2\right).$$

Induction ensures that, for all $t \in \mathbb{N}$,

$$\mathbb{E}\left[\|\boldsymbol{d}_n\|^2\right] \leq \max\left\{\|\boldsymbol{d}_{-1}\|^2, \frac{C_{\mathrm{QHM}}^2}{b} + K_{\mathrm{QHM}}^2\right\} = \frac{C_{\mathrm{QHM}}^2}{b} + K_{\mathrm{QHM}}^2,$$

where $\boldsymbol{d}_{-1} = \boldsymbol{0}$. This completes the proof. $\qquad\square$

### A.6 PROOF OF THEOREM 4.2

*Proof.* Let $\boldsymbol{x} \in \mathbb{R}^d$ and $t \in \mathbb{N}$. The definition of $\boldsymbol{x}_{t+1}$ implies that

$$\|\boldsymbol{x}_{t+1} - \boldsymbol{x}\|^2 = \|(\boldsymbol{x}_t - \eta\boldsymbol{d}_t) - \boldsymbol{x}\|^2$$

$$= \|\boldsymbol{x}_t - \boldsymbol{x}\|^2 - 2\eta\langle \boldsymbol{x}_t - \boldsymbol{x}, \boldsymbol{d}_t \rangle + \eta^2\|\boldsymbol{d}_t\|^2$$

$$= \|\boldsymbol{x}_t - \boldsymbol{x}\|^2 - 2\eta(1 - \nu\beta)\langle \boldsymbol{x}_t - \boldsymbol{x}, \nabla f_{\mathcal{S}_t}(\boldsymbol{x}_t) \rangle + 2\eta\nu\beta\langle \boldsymbol{x} - \boldsymbol{x}_t, \boldsymbol{d}_{t-1} \rangle + \eta^2\|\boldsymbol{d}_t\|^2.$$

Then we have

$$\mathbb{E}_{\boldsymbol{\xi}_t}\left[\langle \boldsymbol{x}_t - \boldsymbol{x}, \nabla f_{\mathcal{S}_t}(\boldsymbol{x}_t)\rangle \Big| \boldsymbol{x}_t\right] = \mathbb{E}_{\boldsymbol{\xi}_t}\left[\frac{1}{2\eta(1-\nu\beta)}\left(\|\boldsymbol{x}_t - \boldsymbol{x}\|^2 - \|\boldsymbol{x}_{t+1} - \boldsymbol{x}\|^2\right)\right.$$

$$\left. + \frac{\nu\beta}{1-\nu\beta}\langle \boldsymbol{x} - \boldsymbol{x}_t, \boldsymbol{d}_{t-1}\rangle + \frac{\eta}{2(1-\nu\beta)}\|\boldsymbol{d}_t\|^2 \Big| \boldsymbol{x}_t\right].$$

On the other hand, Assumptions (A2)(ii) and (A3) guarantee that

$$\mathbb{E}_{\boldsymbol{x}_t}\left[\mathbb{E}_{\boldsymbol{\xi}_t}\left[\langle \boldsymbol{x}_t - \boldsymbol{x}, \nabla f_{\mathcal{S}_t}(\boldsymbol{x}_t)\rangle \Big| \boldsymbol{x}_t\right]\right] = \mathbb{E}_{\boldsymbol{x}_t}\left[\left\langle \boldsymbol{x}_t - \boldsymbol{x}, \mathbb{E}_{\boldsymbol{\xi}_t}\left[\nabla f_{\mathcal{S}_t}(\boldsymbol{x}_t)\Big| \boldsymbol{x}_t\right]\right\rangle\right]$$

$$= \mathbb{E}_{\boldsymbol{x}_t}\left[\langle \boldsymbol{x}_t - \boldsymbol{x}, \nabla f(\boldsymbol{x}_t)\rangle\right].$$

Hence, by taking the total expectation on both sides, we obtain

$$\mathbb{E}\left[\langle \boldsymbol{x}_t - \boldsymbol{x}, \nabla f(\boldsymbol{x}_t)\rangle\right] = \frac{1}{2\eta(1-\nu\beta)}\left(\mathbb{E}\left[\|\boldsymbol{x}_t - \boldsymbol{x}\|^2\right] - \mathbb{E}\left[\|\boldsymbol{x}_{t+1} - \boldsymbol{x}\|^2\right]\right)$$

$$+ \frac{\nu\beta}{1-\nu\beta}\mathbb{E}\left[\langle \boldsymbol{x} - \boldsymbol{x}_t, \boldsymbol{d}_{t-1}\rangle\right] + \frac{\eta}{2(1-\nu\beta)}\mathbb{E}\left[\|\boldsymbol{d}_t\|^2\right].$$

According to Lemma A.5, Assumption 4.1, and the Cauchy-Schwarz inequality,

$$\mathbb{E}\left[\langle \boldsymbol{x}_t - \boldsymbol{x}, \nabla f(\boldsymbol{x}_t)\rangle\right] \le \frac{1}{2\eta(1-\nu\beta)}\left(\mathbb{E}\left[\|\boldsymbol{x}_t - \boldsymbol{x}\|^2\right] - \mathbb{E}\left[\|\boldsymbol{x}_{t+1} - \boldsymbol{x}\|^2\right]\right)$$

$$+ \frac{\nu\beta D(\boldsymbol{x})}{1-\nu\beta}\sqrt{\frac{C_{\mathrm{QHM}}^2}{b} + K_{\mathrm{QHM}}^2} + \frac{\eta}{2(1-\nu\beta)}\left(\frac{C_{\mathrm{QHM}}^2}{b} + K_{\mathrm{QHM}}^2\right).$$

Summing over $t$ from $t = 0$ to $t = T - 1$, we obtain

$$\sum_{t=0}^{T-1}\mathbb{E}\left[\langle \boldsymbol{x}_t - \boldsymbol{x}, \nabla f(\boldsymbol{x}_t)\rangle\right] \le \frac{1}{2\eta(1-\nu\beta)}\left(\mathbb{E}\left[\|\boldsymbol{x}_0 - \boldsymbol{x}\|^2\right] - \mathbb{E}\left[\|\boldsymbol{x}_T - \boldsymbol{x}\|^2\right]\right)$$

$$+ \frac{\nu\beta D(\boldsymbol{x})}{1-\nu\beta}\sqrt{\frac{C_{\mathrm{QHM}}^2}{b} + K_{\mathrm{QHM}}^2}T + \frac{\eta}{2(1-\nu\beta)}\left(\frac{C_{\mathrm{QHM}}^2}{b} + K_{\mathrm{QHM}}^2\right)T.$$

Therefore,

$$\frac{1}{T}\sum_{t=0}^{T-1}\mathbb{E}\left[\langle \boldsymbol{x}_t - \boldsymbol{x}, \nabla f(\boldsymbol{x}_t)\rangle\right] \le \frac{\|\boldsymbol{x}_0 - \boldsymbol{x}\|^2}{2\eta(1-\nu\beta)T} + \frac{\nu\beta D(\boldsymbol{x})}{1-\nu\beta}\sqrt{\frac{C_{\mathrm{QHM}}^2}{b} + K_{\mathrm{QHM}}^2}$$

$$+ \frac{\eta}{2(1-\nu\beta)}\left(\frac{C_{\mathrm{QHM}}^2}{b} + K_{\mathrm{QHM}}^2\right).$$

This completes the proof. $\qquad\square$

### A.7 Convergence analysis of SGD

convergence analysis of SGD is needed to discuss critical batch size.

**Theorem A.1.** *Suppose that Assumptions (A1)-(A4) hold and consider the sequence $(\boldsymbol{x}_t)_{t\in\mathbb{N}}$ generated by SGD. Then, for all $\boldsymbol{x} \in \mathbb{R}^d$ and all $T \ge 1$, the following holds:*

$$\frac{1}{T}\sum_{t=0}^{T-1}\mathbb{E}\left[\langle \boldsymbol{x}_t - \boldsymbol{x}, \nabla f(\boldsymbol{x}_t)\rangle\right] \le \frac{\|\boldsymbol{x}_0 - \boldsymbol{x}\|^2}{2\eta T} + \frac{\eta}{2}\left(\frac{C_{\mathrm{SGD}}^2}{b} + K_{\mathrm{SGD}}^2\right).$$

*Proof.* Let $\boldsymbol{x} \in \mathbb{R}^d$ and $t \in \mathbb{N}$. The definition of $\boldsymbol{x}_{t+1}$ implies that

$$\|\boldsymbol{x}_{t+1} - \boldsymbol{x}\|^2 = \|(\boldsymbol{x}_t - \eta\nabla f_{\mathcal{S}_t}(\boldsymbol{x}_t)) - \boldsymbol{x}\|^2$$

$$= \|\boldsymbol{x}_t - \boldsymbol{x}\|^2 - 2\eta\langle \boldsymbol{x}_t - \boldsymbol{x}, \nabla f_{\mathcal{S}_t}(\boldsymbol{x}_t)\rangle + \eta^2\|\nabla f_{\mathcal{S}_t}(\boldsymbol{x}_t)\|^2.$$

Then we have

$$
\mathbb{E}_{\boldsymbol{\xi}_t} \left[ \langle \boldsymbol{x}_t - \boldsymbol{x}, \nabla f_{\mathcal{S}_t}(\boldsymbol{x}_t) \rangle \Big| \boldsymbol{x}_t \right] = \mathbb{E}_{\boldsymbol{\xi}_t} \left[ \frac{1}{2\eta} \left( \|\boldsymbol{x}_t - \boldsymbol{x}\|^2 - \|\boldsymbol{x}_{t+1} - \boldsymbol{x}\|^2 \right) + \frac{\eta}{2} \|\nabla f_{\mathcal{S}_t}(\boldsymbol{x}_t)\|^2 \Big| \boldsymbol{x}_t \right].
$$

On the other hand, Assumptions (A2)(ii) and (A3) guarantee that

$$
\mathbb{E}_{\boldsymbol{x}_t} \left[ \mathbb{E}_{\boldsymbol{\xi}_t} \left[ \langle \boldsymbol{x}_t - \boldsymbol{x}, \nabla f_{\mathcal{S}_t}(\boldsymbol{x}_t) \rangle \Big| \boldsymbol{x}_t \right] \right] = \mathbb{E}_{\boldsymbol{x}_t} \left[ \left\langle \boldsymbol{x}_t - \boldsymbol{x}, \mathbb{E}_{\boldsymbol{\xi}_t} \left[ \nabla f_{\mathcal{S}_t}(\boldsymbol{x}_t) \Big| \boldsymbol{x}_t \right] \right\rangle \right]
$$
$$
= \mathbb{E}_{\boldsymbol{x}_t} \left[ \langle \boldsymbol{x}_t - \boldsymbol{x}, \nabla f(\boldsymbol{x}_t) \rangle \right].
$$

Hence, by taking the total expectation on both sides, we obtain

$$
\mathbb{E} \left[ \langle \boldsymbol{x}_t - \boldsymbol{x}, \nabla f(\boldsymbol{x}_t) \rangle \right] = \frac{1}{2\eta} \left( \mathbb{E} \left[ \|\boldsymbol{x}_t - \boldsymbol{x}\|^2 \right] - \mathbb{E} \left[ \|\boldsymbol{x}_{t+1} - \boldsymbol{x}\|^2 \right] \right) + \frac{\eta}{2} \mathbb{E} \left[ \|\nabla f_{\mathcal{S}_t}(\boldsymbol{x}_t)\|^2 \right].
$$

According to Lemma A.2,

$$
\mathbb{E} \left[ \langle \boldsymbol{x}_t - \boldsymbol{x}, \nabla f(\boldsymbol{x}_t) \rangle \right] \leq \frac{1}{2\eta} \left( \mathbb{E} \left[ \|\boldsymbol{x}_t - \boldsymbol{x}\|^2 \right] - \mathbb{E} \left[ \|\boldsymbol{x}_{t+1} - \boldsymbol{x}\|^2 \right] \right) + \frac{\eta}{2} \left( \frac{C_{\text{SGD}}^2}{b} + K_{\text{SGD}}^2 \right).
$$

Summing over $t$ from $t = 0$ to $t = T - 1$, we obtain

$$
\sum_{t=0}^{T-1} \mathbb{E} \left[ \langle \boldsymbol{x}_t - \boldsymbol{x}, \nabla f(\boldsymbol{x}_t) \rangle \right] \leq \frac{1}{2\eta} \left( \mathbb{E} \left[ \|\boldsymbol{x}_0 - \boldsymbol{x}\|^2 \right] - \mathbb{E} \left[ \|\boldsymbol{x}_T - \boldsymbol{x}\|^2 \right] \right) + \frac{\eta}{2} \left( \frac{C_{\text{SGD}}^2}{b} + K_{\text{SGD}}^2 \right) T
$$

Therefore,

$$
\frac{1}{T} \sum_{t=0}^{T-1} \mathbb{E} \left[ \langle \boldsymbol{x}_t - \boldsymbol{x}, \nabla f(\boldsymbol{x}_t) \rangle \right] \leq \frac{\|\boldsymbol{x}_0 - \boldsymbol{x}\|^2}{2\eta T} + \frac{\eta}{2} \left( \frac{C_{\text{SGD}}^2}{b} + K_{\text{SGD}}^2 \right).
$$

This completes the proof. □

# B  ANALYSIS OF CRITICAL BATCH SIZE FOR SGD, SHB, AND QHM

Following earlier studies (Iiduka, 2022b; Sato & Iiduka, 2023a), we derive Proposition 4.1 for estimating a lower bound on the critical batch size. First, the convergence of the optimizer must be analyzed (Theorems A.1, 4.1, and 4.2), and on the basis of that analysis, the number of steps $T$ required for training is defined as a function of batch size $b$ (Theorem B.1). Next, computational complexity is expressed as the number of steps multiplied by the batch size, and computational complexity $T(b)b$ is defined as a function of batch size $b$. Finally, we identify critical batch size $b^\star$ that minimizes computational complexity function $T(b)b$ (Theorem B.2) and transform the lower bound for each optimizer (Proposition 4.1).

## B.1  RELATIONSHIP BETWEEN BATCH SIZE AND NUMBER OF STEPS NEEDED FOR $\epsilon$-APPROXIMATION

According to Theorems A.1, 4.1, and 4.2, the following hold:

(i) for SGD,

$$
\frac{1}{T} \sum_{t=0}^{T-1} \mathbb{E} \left[ \langle \boldsymbol{x}_t - \boldsymbol{x}, \nabla f(\boldsymbol{x}_t) \rangle \right] \leq \frac{\|\boldsymbol{x}_0 - \boldsymbol{x}\|^2}{2\eta T} + \frac{\eta}{2} \left( \frac{C_{\text{SGD}}^2}{b} + K_{\text{SGD}}^2 \right)
$$
$$
= \underbrace{\frac{\|\boldsymbol{x}_0 - \boldsymbol{x}\|^2}{2\eta}}_{=:X_{\text{SGD}}} \frac{1}{T} + \underbrace{\frac{\eta C_{\text{SGD}}^2}{2}}_{=:Y_{\text{SGD}}} \frac{1}{b} + \underbrace{\frac{\eta K_{\text{SGD}}^2}{2}}_{=:Z_{\text{SGD}}}; \tag{10}
$$

(ii) for SHB,

$$
\frac{1}{T} \sum_{t=0}^{T-1} \mathbb{E}\left[\langle \boldsymbol{x}_t - \boldsymbol{x}, \nabla f(\boldsymbol{x}_t)\rangle\right]
$$

$$
\leq \frac{\|\boldsymbol{x}_0 - \boldsymbol{x}\|^2}{2\eta T} + \frac{\beta D(\boldsymbol{x})}{1-\beta} \sqrt{C_{\text{SHB}}^2 + K_{\text{SHB}}^2} + \frac{\eta\left(\beta^2 - \beta + 1\right)}{2\beta(1-\beta)^2}\left(\frac{C_{\text{SHB}}^2}{b} + K_{\text{SHB}}^2\right)
$$

$$
= \underbrace{\frac{\|\boldsymbol{x}_0 - \boldsymbol{x}\|^2}{2\eta}}_{=:X_{\text{SHB}}} \frac{1}{T} + \underbrace{\frac{\eta\left(\beta^2 - \beta + 1\right) C_{\text{SHB}}^2}{2\beta(1-\beta)^2}}_{=:Y_{\text{SHB}}} \frac{1}{b}
$$

$$
+ \underbrace{\left\{\frac{\eta\left(\beta^2 - \beta + 1\right)}{2\beta(1-\beta)^2} K_{\text{SHB}}^2 + \frac{\beta D(\boldsymbol{x})}{1-\beta} \sqrt{C_{\text{SHB}}^2 + K_{\text{SHB}}^2}\right\}}_{=:Z_{\text{SHB}}}; \tag{11}
$$

(iii) for QHM,

$$
\frac{1}{T} \sum_{t=0}^{T-1} \mathbb{E}\left[\langle \boldsymbol{x}_t - \boldsymbol{x}, \nabla f(\boldsymbol{x}_t)\rangle\right]
$$

$$
\leq \frac{\|\boldsymbol{x}_0 - \boldsymbol{x}\|^2}{2\eta(1-\nu\beta)T} + \frac{\nu\beta D(\boldsymbol{x})}{1-\nu\beta} \sqrt{C_{\text{QHM}}^2 + K_{\text{QHM}}^2} + \frac{\eta}{2(1-\nu\beta)}\left(\frac{C_{\text{QHM}}^2}{b} + K_{\text{QHM}}^2\right)
$$

$$
= \underbrace{\frac{\|\boldsymbol{x}_0 - \boldsymbol{x}\|^2}{2\eta(1-\nu\beta)}}_{=:X_{\text{QHM}}} \frac{1}{T} + \underbrace{\frac{\eta C_{\text{QHM}}^2}{2(1-\nu\beta)}}_{=:Y_{\text{QHM}}} \frac{1}{b} + \underbrace{\left\{\frac{\eta C_{\text{QHM}}^2}{2(1-\nu\beta)} K_{\text{QHM}}^2 + \frac{\nu\beta D(\boldsymbol{x})}{1-\nu\beta} \sqrt{C_{\text{QHM}}^2 + K_{\text{QHM}}^2}\right\}}_{=:Z_{\text{QHM}}}.
$$

$$\tag{12}$$

The relationship between $b$ and number of steps $T_{\text{SGD}}$, $T_{\text{SHB}}$, and $T_{\text{QHM}}$ satisfying an $\epsilon$-approximation is as follows:

**Theorem B.1.** *Suppose that Assumptions (A1)-(A4), and 4.1 hold and consider SGD, SHB, and QHM. Then, $T_{\text{SGD}}(b)$, $T_{\text{SHB}}(b)$, and $T_{\text{QHM}}(b)$ defined by*

$$
T_{\text{SGD}}(b) := \frac{X_{SGD}b}{(\epsilon^2 - Z_{\text{SGD}})b - Y_{\text{SGD}}} \leq T_{\text{SGD}} \text{ for } b > \frac{Y_{\text{SGD}}}{\epsilon^2 - Z_{\text{SGD}}}, \tag{13}
$$

$$
T_{\text{SHB}}(b) := \frac{X_{\text{SHB}}b}{(\epsilon^2 - Z_{\text{SHB}})b - Y_{\text{SHB}}} \leq T_{\text{SHB}} \text{ for } b > \frac{Y_{\text{SHB}}}{\epsilon^2 - Z_{\text{SHB}}}, \tag{14}
$$

$$
T_{\text{QHM}}(b) := \frac{X_{\text{QHM}}b}{(\epsilon^2 - Z_{\text{QHM}})b - Y_{\text{QHM}}} \leq T_{\text{QHM}} \text{ for } b > \frac{Y_{\text{QHM}}}{\epsilon^2 - Z_{\text{QHM}}} \tag{15}
$$

*satisfy*

$$
\frac{1}{T_{\text{SGD}}} \sum_{t=0}^{T_{\text{SGD}}-1} \mathbb{E}\left[\langle \boldsymbol{x}_t - \boldsymbol{x}, \nabla f(\boldsymbol{x}_t)\rangle\right] \leq \epsilon^2,
$$

$$
\frac{1}{T_{\text{SHB}}} \sum_{t=0}^{T_{\text{SHB}}-1} \mathbb{E}\left[\langle \boldsymbol{x}_t - \boldsymbol{x}, \nabla f(\boldsymbol{x}_t)\rangle\right] \leq \epsilon^2,
$$

$$
\frac{1}{T_{\text{QHM}}} \sum_{t=0}^{T_{\text{QHM}}-1} \mathbb{E}\left[\langle \boldsymbol{x}_t - \boldsymbol{x}, \nabla f(\boldsymbol{x}_t)\rangle\right] \leq \epsilon^2.
$$

*In addition, the functions $T_{\text{SGD}}(b)$, $T_{\text{SHB}}(b)$, and $T_{\text{QHM}}(b)$ defined by (13)-(15) are monotone decreasing and convex for $b > \frac{Y_{\text{SGD}}}{\epsilon^2 - Z_{\text{SGD}}}$, $b > \frac{Y_{\text{SHB}}}{\epsilon^2 - Z_{\text{SHB}}}$, and $b > \frac{Y_{\text{QHM}}}{\epsilon^2 - Z_{\text{QHM}}}$.*

*Proof.* According to (10) and (13), SGD achieves an $\epsilon$-approximation. We have that, for $b > \frac{Y_{\text{SGD}}}{\epsilon^2 - Z_{\text{SGD}}}$,

$$\frac{\mathrm{d}T_{\text{SGD}}(b)}{\mathrm{d}b} = \frac{-X_{\text{SGD}}Y_{\text{SGD}}}{\left\{(\epsilon^2 - Z_{\text{SGD}})b - Y_{\text{SGD}}\right\}^2} \leq 0,$$

$$\frac{\mathrm{d}^2 T_{\text{SGD}}(b)}{\mathrm{d}b^2} = \frac{2X_{\text{SGD}}Y_{\text{SGD}}(\epsilon^2 - Z_{\text{SGD}})}{\left\{(\epsilon^2 - Z_{\text{SGD}})b - Y_{\text{SGD}}\right\}^3} \geq 0.$$

Therefore, $T_{\text{SGD}}(b)$ is monotone decreasing and convex for $b > \frac{Y_{\text{SGD}}}{\epsilon^2 - Z_{\text{SGD}}}$. The discussions for SHB and QHM are similar to the one for SGD. This completes the proof. $\quad\square$

### B.2 EXISTENCE OF A CRITICAL BATCH SIZE

The critical batch size minimizes the computational complexity for training. Here, we use stochastic first-order oracle (SFO) complexity as a measure of computational complexity. Since the stochastic gradient is computed $b$ times per step, SFO complexity is defined as

$$T_{\text{SGD}}(b)b = \frac{X_{\text{SGD}}b^2}{(\epsilon^2 - Z_{\text{SGD}})b - Y_{\text{SGD}}},$$

$$T_{\text{SHB}}(b)b = \frac{X_{\text{SHB}}b^2}{(\epsilon^2 - Z_{\text{SHB}})b - Y_{\text{SHB}}}, \text{ and} \tag{16}$$

$$T_{\text{QHM}}(b)b = \frac{X_{\text{QHM}}b^2}{(\epsilon^2 - Z_{\text{QHM}})b - Y_{\text{QHM}}}.$$

The following theorem guarantees the existence of critical batch sizes that are global minimizers of $T_{\text{SGD}}(b)b$, $T_{\text{SHB}}(b)b$, and $T_{\text{QHM}}(b)b$ defined by (16).

**Theorem B.2.** *Suppose that Assumptions (A1)-(A4) and 4.1 hold and consider SGD, SHB, and QHM. Then, there exist*

$$b_{\text{SGD}}^\star := \frac{2Y_{\text{SGD}}}{\epsilon^2 - Z_{\text{SGD}}}, b_{\text{SHB}}^\star := \frac{2Y_{\text{SHB}}}{\epsilon^2 - Z_{\text{SHB}}}, \text{ and } b_{\text{QHM}}^\star := \frac{2Y_{\text{QHM}}}{\epsilon^2 - Z_{\text{QHM}}} \tag{17}$$

*such that $b_{\text{SGD}}^\star$ minimizes the convex function $T_{\text{SGD}}(b)b$ $(b > Y_{\text{SGD}}/(\epsilon^2 - Z_{\text{SGD}}))$, $b_{\text{SHB}}^\star$ minimizes the convex function $T_{\text{SHB}}(b)b$ $(b > Y_{\text{SHB}}/(\epsilon^2 - Z_{\text{SHB}}))$, and $b_{\text{QHM}}^\star$ minimizes the convex function $T_{\text{QHM}}(b)b$ $(b > Y_{\text{QHM}}/(\epsilon^2 - Z_{\text{QHM}}))$.*

*Proof.* From (17), we have that, for $b > Y_{\text{SGD}}/(\epsilon^2 - Z_{\text{SGD}}))$,

$$\frac{\mathrm{d}T_{\text{SGD}}(b)b}{\mathrm{d}b} = \frac{X_{\text{SGD}}b\left\{(\epsilon^2 - Z_{\text{SGD}})b - 2Y_{\text{SGD}}\right\}}{\left\{(\epsilon^2 - Z_{\text{SGD}})b - Y_{\text{SGD}}\right\}^2},$$

$$\frac{\mathrm{d}^2 T_{\text{SGD}}(b)b}{\mathrm{d}b^2} = \frac{2X_{\text{SGD}}Y_{\text{SGD}}^2}{\left\{(\epsilon^2 - Z_{\text{SGD}})b - Y_{\text{SGD}}\right\}^3} \geq 0.$$

Hence, $T_{\text{SGD}}(b)b$ is convex for $b > Y_{\text{SGD}}/(\epsilon^2 - Z_{\text{SGD}})$ and

$$\frac{\mathrm{d}T_{\text{SGD}}(b)b}{\mathrm{d}b} \begin{cases} < 0 & \text{if } b < b_{\text{SGD}}^\star, \\ = 0 & \text{if } b = b_{\text{SGD}}^\star = \frac{2Y_{\text{SGD}}}{\epsilon^2 - Z_{\text{SGD}}}, \\ > 0 & \text{if } b > b_{\text{SGD}}^\star. \end{cases}$$

The discussions for SHB and QHM are similar to the one for SGD. This completes the proof. $\quad\square$

### B.3 PROOF OF PROPOSITION 4.1

*Proof.* Theorem B.2 and the definition of $Y_{\text{SGD}}$ and $Z_{\text{SGD}}$ (see (10)) ensure that

$$b_{\text{SGD}}^\star := \frac{2Y_{\text{SGD}}}{\epsilon^2 - Z_{\text{SGD}}} > \frac{2Y_{\text{SGD}}}{\epsilon^2} = \frac{2}{\epsilon^2} \cdot \frac{\eta C_{\text{SGD}}^2}{2} = \frac{\eta C_{\text{SGD}}^2}{\epsilon^2}.$$

Similarly, for SHB, from Theorem B.2 and the definition of $Y_{\text{SHB}}$ and $Z_{\text{SHB}}$ (see (11)), we obtain

$$b_{\text{SHB}}^\star := \frac{2Y_{\text{SHB}}}{\epsilon^2 - Z_{\text{SHB}}} > \frac{2Y_{\text{SHB}}}{\epsilon^2} = \frac{2}{\epsilon^2} \cdot \frac{\eta\left(\beta^2 - \beta + 1\right)C_{\text{SHB}}^2}{2\beta(1-\beta)^2} = \frac{\eta\left(\beta^2 - \beta + 1\right)C_{\text{SHB}}^2}{\beta(1-\beta)^2\epsilon^2}.$$

Finally, for QHM, from Theorem B.2 and the definition of $Y_{\text{QHM}}$ and $Z_{\text{QHM}}$ (see (12)), we obtain

$$b_{\text{QHM}}^\star := \frac{2Y_{\text{QHM}}}{\epsilon^2 - Z_{\text{QHM}}} > \frac{2Y_{\text{QHM}}}{\epsilon^2} = \frac{2}{\epsilon^2} \cdot \frac{\eta C_{\text{QHM}}^2}{2(1-\nu\beta)} = \frac{\eta C_{\text{QHM}}^2}{(1-\nu\beta)\epsilon^2}.$$

This completes the proof. $\qquad\square$

### B.4 MORE DETAILS ON EXPERIMENTAL RESULTS IN SECTION 4.3

Since SFO complexity is expressed as the product of the number of steps and the batch size, we first measured the number of steps $T$ required to achieve a sufficiently small gradient norm for each batch size. Figure 6 plots the number of steps $T$ needed to achieve the gradient norm of the past $t$ steps at time $t$ to average less than $\epsilon = 0.5$ versus batch size $b$. The figure shows that the number of steps for each optimizer was mostly monotone decreasing and convex with respect to batch size $b$, which provides experimental support for Theorem B.1. Next, we calculated SFO complexity by multiplying number of steps $T$ by batch size $b$. As shown in Figure 7, SFO complexity for each optimizer was convex with respect to batch size $b$, which provides experimental support for Theorem B.2. We performed similar experiments on training WideResNet-28-10 on CIFAR100 and obtained similar results. The results are plotted in Figures 12 and 13.

### B.5 COMPUTING VARIANCE OF STOCHASTIC GRADIENT USING PROPOSITION 4.1

**Training ResNet18 on CIFAR100 dataset:** From Proposition 4.1 and the hyperparameters used in the experiments for training ResNet18 on the CIFAR100 dataset, we obtained

$$C_{\text{SGD}}^2 < \frac{b_{\text{SGD}}^\star \epsilon^2}{\eta} = \frac{2^9 \cdot (0.5)^2}{0.1} = 1280,$$

$$C_{\text{SHB}}^2 < \frac{b_{\text{SHB}}^\star \epsilon^2 \beta(1-\beta)^2}{\eta(\beta^2 - \beta + 1)} = \frac{2^{10} \cdot (0.5)^2 \cdot 0.9 \cdot (0.1)^2}{0.1 \cdot 0.91} = 25.318,$$

$$C_{\text{NSHB}}^2 < \frac{b_{\text{NSHB}}^\star \epsilon^2 (1-\nu\beta)}{\eta} = \frac{2^9 \cdot (0.5)^2 \cdot (1 - 1 \cdot 0.9)}{0.1} = 128,$$

where $\eta = 0.1, \beta = 0.9, \nu = 1$, and $\epsilon = 0.5$ were used in the experiments and $b_{\text{SGD}}^\star = 2^9, b_{\text{SHB}}^\star = 2^{10}$, and $b_{\text{NSHB}}^\star = 2^9$ were measured by experiment (see Figure 7).

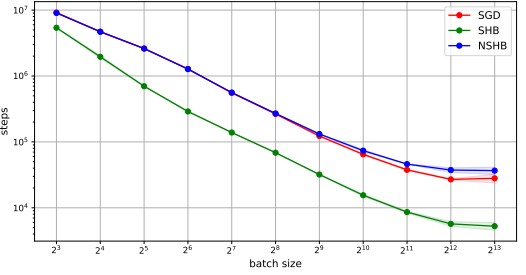
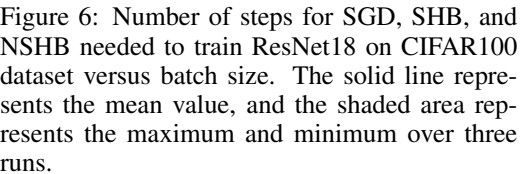
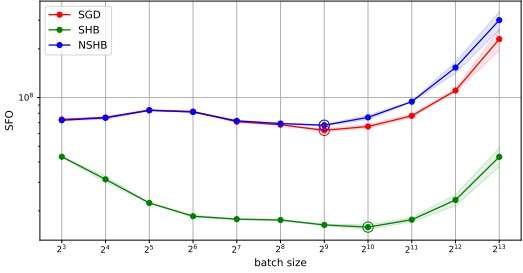

Figure 6: Number of steps for SGD, SHB, and NSHB needed to train ResNet18 on CIFAR100 dataset versus batch size. The solid line represents the mean value, and the shaded area represents the maximum and minimum over three runs.

Figure 7: SFO complexities for SGD, SHB, and NSHB needed to train ResNet18 on CIFAR100 dataset versus batch size. The double circle denotes the critical batch size that minimizes SFO complexity. The solid line represents the mean value, and the shaded area represents the maximum and minimum over three runs. **This is the same graph shown in Figure 2.**

To discuss the noise level of smoothing in Section 3, we also measured the gradient norm and its upper bound. The measured gradient norm was larger for smaller batch sizes, with maximum values of $4.528$, $1.77$, and $4.5$ for SGD, SHB, and NSHB, respectively. We used this value as an upper bound on the gradient norm (i.e., $K_{\text{SGD}} := 4.528$, $K_{\text{SHB}} := 1.77$, and $K_{\text{NSHB}} := 4.5$) for training ResNet18 on the CIFAR100 dataset.

**Training WideResNet-28-10 on CIFAR100 dataset:** From a similar discussion, for training WideResNet-28-10 on the CIFAR100 dataset, we obtained

$$C_{\text{SGD}}^2 < \frac{b_{\text{SGD}}^\star \epsilon^2}{\eta} = \frac{2^2 \cdot (0.5)^2}{0.1} = 10,$$

$$C_{\text{SHB}}^2 < \frac{b_{\text{SHB}}^\star \epsilon^2 \beta(1-\beta)^2}{\eta(\beta^2 - \beta + 1)} = \frac{2^5 \cdot (0.5)^2 \cdot 0.9 \cdot (0.1)^2}{0.1 \cdot 0.91} = 0.79,$$

$$C_{\text{NSHB}}^2 < \frac{b_{\text{NSHB}}^\star \epsilon^2 (1 - \nu\beta)}{\eta} = \frac{2^2 \cdot (0.5)^2 \cdot (1 - 1 \cdot 0.9)}{0.1} = 1.0,$$

where $\eta = 0.1, \beta = 0.9, \nu = 1$, and $\epsilon = 0.5$ were used in the experiments and $b_{\text{SGD}}^\star = 2^2, b_{\text{SHB}}^\star = 2^5$, and $b_{\text{NSHB}}^\star = 2^2$ were measured by experiment (see Figure 13). We also used it as an upper bound on the gradient norm ($K_{\text{SGD}} := 4.259$, $K_{\text{SHB}} := 1.66$, and $K_{\text{NSHB}} := 4.262$) for training WideResNet-28-10 on the CIFAR100 dataset.

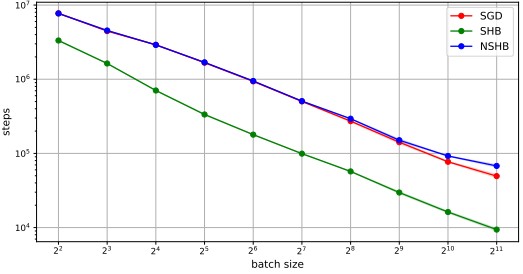 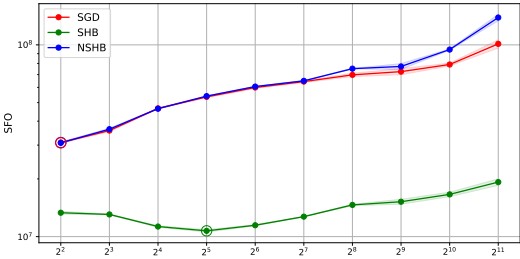

Figure 8: Number of steps for SGD, SHB, and NSHB needed to train WideResNet-28-10 on CI-FAR100 dataset versus batch size. The solid line represents the mean value, and the shaded area represents the maximum and minimum over three runs.

Figure 9: SFO complexities for SGD, SHB, and NSHB needed to train WideResNet-28-10 on CI-FAR100 dataset versus batch size. The double circle denotes the critical batch size that minimizes SFO complexity. The solid line represents the mean value, and the shaded area represents the maximum and minimum over three runs.

**Training MobileNetV2 on CIFAR100 dataset:** From a similar discussion, for training MobileNet-v2 on the CIFAR100 dataset, we obtained

$$C_{\text{SGD}}^2 < \frac{b_{\text{SGD}}^\star \epsilon^2}{\eta} = \frac{2^3 \cdot (0.5)^2}{0.1} = 20,$$

$$C_{\text{SHB}}^2 < \frac{b_{\text{SHB}}^\star \epsilon^2 \beta(1-\beta)^2}{\eta(\beta^2 - \beta + 1)} = \frac{2^8 \cdot (0.5)^2 \cdot 0.9 \cdot (0.1)^2}{0.1 \cdot 0.91} = 6.33,$$

$$C_{\text{NSHB}}^2 < \frac{b_{\text{NSHB}}^\star \epsilon^2 (1 - \nu\beta)}{\eta} = \frac{2^3 \cdot (0.5)^2 \cdot (1 - 1 \cdot 0.9)}{0.1} = 2,$$

where $\eta = 0.1, \beta = 0.9, \nu = 1$, and $\epsilon = 0.5$ were used in the experiments and $b_{\text{SGD}}^\star = 2^2, b_{\text{SHB}}^\star = 2^5$, and $b_{\text{NSHB}}^\star = 2^2$ were measured by experiment (see Figure 13). We also measured the gradient norm and its upper bound; the maximum value of 1.43 for SHB, i.e., $K_{\text{SHB}} := 1.43$.

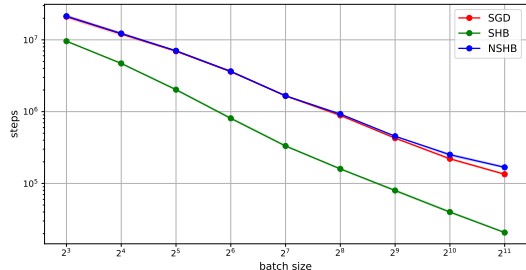 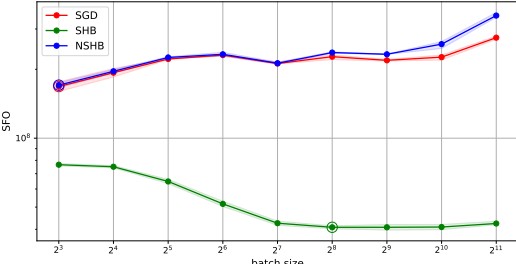

Figure 10: Number of steps for SGD, SHB, and NSHB needed to train MobileNetV2 on CIFAR100 dataset versus batch size. The solid line represents the mean value, and the shaded area represents the maximum and minimum over three runs.

Figure 11: SFO complexities for SGD, SHB, and NSHB needed to train MovileNetV2 on CIFAR100 dataset versus batch size. The double circle denotes the critical batch size that minimizes SFO complexity. The solid line represents the mean value, and the shaded area represents the maximum and minimum over three runs.

**Training ResNet18 on CIFAR10 dataset:** From a similar discussion, for training ResNet18 on the CIFAR10 dataset, we obtained

$$C_{\text{SGD}}^2 < \frac{b_{\text{SGD}}^\star \epsilon^2}{\eta} = \frac{2^3 \cdot (0.5)^2}{0.1} = 20,$$

$$C_{\text{SHB}}^2 < \frac{b_{\text{SHB}}^\star \epsilon^2 \beta (1 - \beta)^2}{\eta(\beta^2 - \beta + 1)} = \frac{2^5 \cdot (0.5)^2 \cdot 0.9 \cdot (0.1)^2}{0.1 \cdot 0.91} = 0.79,$$

$$C_{\text{NSHB}}^2 < \frac{b_{\text{NSHB}}^\star \epsilon^2 (1 - \nu\beta)}{\eta} = \frac{2^3 \cdot (0.5)^2 \cdot (1 - 1 \cdot 0.9)}{0.1} = 2,$$

where $\eta = 0.1, \beta = 0.9, \nu = 1$, and $\epsilon = 0.5$ were used in the experiments and $b_{\text{SGD}}^\star = 2^2, b_{\text{SHB}}^\star = 2^5$, and $b_{\text{NSHB}}^\star = 2^2$ were measured by experiment (see Figure 13). We also measured the gradient norm and its upper bound; the maximum value of 1.134 for SHB, i.e., $K_{\text{SHB}} := 1.134$.

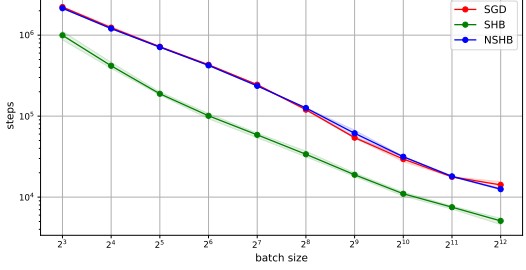 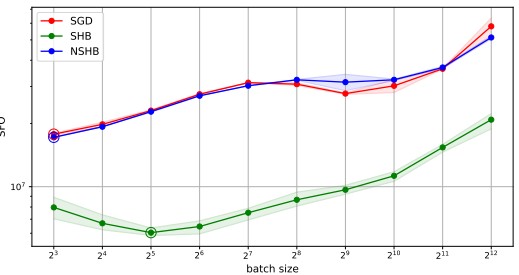

Figure 12: Number of steps for SGD, SHB, and NSHB needed to train ResNet18 on CIFAR10 dataset versus batch size. The solid line represents the mean value, and the shaded area represents the maximum and minimum over three runs.

Figure 13: SFO complexities for SGD, SHB, and NSHB needed to train ResNet18 on CIFAR10 dataset versus batch size. The double circle denotes the critical batch size that minimizes SFO complexity. The solid line represents the mean value, and the shaded area represents the maximum and minimum over three runs.

## C SMOOTHING PROPERTY OF OPTIMIZERS WITH A MINI-BATCH STOCHASTIC GRADIENT

### C.1 PROOF OF LEMMA 2.1

*Proof.* From Definition 2.1 and (C2), we have, for all $\boldsymbol{x}, \boldsymbol{y} \in \mathbb{R}^d$,

$$
\begin{aligned}
\left|\hat{f}_\delta(\boldsymbol{x}) - f(\boldsymbol{x})\right| &= |\mathbb{E}_{\boldsymbol{u}}\left[f(\boldsymbol{x} - \delta\boldsymbol{u})\right] - f(\boldsymbol{x})| \\
&= |\mathbb{E}_{\boldsymbol{u}}\left[f(\boldsymbol{x} - \delta\boldsymbol{u}) - f(\boldsymbol{x})\right]| \\
&\le \mathbb{E}_{\boldsymbol{u}}\left[|f(\boldsymbol{x} - \delta\boldsymbol{u}) - f(\boldsymbol{x})|\right] \\
&\le \mathbb{E}_{\boldsymbol{u}}\left[L_f\|(\boldsymbol{x} - \delta\boldsymbol{u}) - \boldsymbol{x}\|\right] \\
&= \delta L_f \mathbb{E}_{\boldsymbol{u}}\left[\|\boldsymbol{u}\|\right].
\end{aligned}
$$

This completes the proof. □

**Remark C.1.** Since the standard normal distribution in high dimensions $d$ is close to a uniform distribution on a sphere of radius $\sqrt{d}$ (Vershynin, 2018, Section 3.3.3), in deep neural network training, for all $\boldsymbol{u} \sim \mathcal{N}\left(\boldsymbol{0}; \frac{1}{\sqrt{d}}I_d\right)$,

$$
\|\boldsymbol{u}\| \approx 1.
$$

Therefore, we have

$$
\left|\hat{f}_\delta(\boldsymbol{x}) - f(\boldsymbol{x})\right| \le \delta L_f.
$$

### C.2 PROOF OF THEOREM 3.1

*Proof.* The definition of $\boldsymbol{m}_t$ implies that

$$
\begin{aligned}
\left\|\boldsymbol{\omega}_t^{\mathrm{SHB}}\right\|^2 &= \|\boldsymbol{m}_t - \nabla f(\boldsymbol{x}_t)\|^2 \\
&= \|\nabla f_{\mathcal{S}_t}(\boldsymbol{x}_t) + \beta\boldsymbol{m}_{t-1} - \nabla f(\boldsymbol{x}_t)\|^2 \\
&= \|\nabla f_{\mathcal{S}_t}(\boldsymbol{x}_t) - \nabla f(\boldsymbol{x}_t)\|^2 + 2\beta\langle\nabla f_{\mathcal{S}_t}(\boldsymbol{x}_t) - \nabla f(\boldsymbol{x}_t), \boldsymbol{m}_{t-1}\rangle + \beta^2\|\boldsymbol{m}_{t-1}\|^2.
\end{aligned}
$$

Hence, from Lemmas A.2 and A.4, we obtain

$$
\begin{aligned}
\mathbb{E}\left[\left\|\boldsymbol{\omega}_t^{\mathrm{SHB}}\right\|^2\right] &= \mathbb{E}\left[\|\nabla f_{\mathcal{S}_t}(\boldsymbol{x}_t) - \nabla f(\boldsymbol{x}_t)\|^2\right] + \beta^2\mathbb{E}\left[\|\boldsymbol{m}_{t-1}\|^2\right] \\
&\le \frac{C_{\mathrm{SHB}}^2}{b} + \frac{\beta(\beta^2 - \beta + 1)}{(1 - \beta)^2}\left(\frac{C_{\mathrm{SHB}}^2}{b} + K_{\mathrm{SHB}}^2\right) \\
&= \left(1 + \frac{\beta(\beta^2 - \beta + 1)}{(1 - \beta)^2}\right)\frac{C_{\mathrm{SHB}}^2}{b} + \frac{\beta(\beta^2 - \beta + 1)}{(1 - \beta)^2}K_{\mathrm{SHB}}^2.
\end{aligned}
$$

Similarly, the definition of $\boldsymbol{d}_t$ implies that

$$
\begin{aligned}
\left\|\boldsymbol{\omega}_t^{\mathrm{QHM}}\right\|^2 &= \|\boldsymbol{d}_t - \nabla f(\boldsymbol{x}_t)\|^2 \\
&= \|(1 - \nu\beta)\nabla f_{\mathcal{S}_t}(\boldsymbol{x}_t) + \nu\beta\boldsymbol{d}_{t-1} - \nabla f(\boldsymbol{x}_t)\|^2 \\
&= \|(1 - \nu\beta)\left(\nabla f_{\mathcal{S}_t}(\boldsymbol{x}_t) - \nabla f(\boldsymbol{x}_t)\right) + \nu\beta(\boldsymbol{d}_{t-1} - \nabla f(\boldsymbol{x}_t))\|^2 \\
&= (1 - \nu\beta)^2\|\nabla f_{\mathcal{S}_t}(\boldsymbol{x}_t) - \nabla f(\boldsymbol{x}_t)\|^2 + \nu^2\beta^2\|\boldsymbol{d}_{t-1} - \nabla f(\boldsymbol{x}_t)\|^2 \\
&\quad + 2\nu\beta(1 - \nu\beta)\langle\nabla f_{\mathcal{S}_t}(\boldsymbol{x}_t) - \nabla f(\boldsymbol{x}_t), \boldsymbol{d}_{t-1} - \nabla f(\boldsymbol{x}_t)\rangle.
\end{aligned}
$$

Therefore, from Assumption (A2)(i) and $\nu\beta < 1$, we obtain

$$
\mathbb{E}\left[\left\|\boldsymbol{\omega}_t^{\mathrm{QHM}}\right\|^2\right] = (1 - \nu\beta)^2\mathbb{E}\left[\|\nabla f_{\mathcal{S}_t}(\boldsymbol{x}_t) - \nabla f(\boldsymbol{x}_t)\|^2\right] + \nu^2\beta^2\mathbb{E}\left[\|\boldsymbol{d}_{t-1} - \nabla f(\boldsymbol{x}_t)\|^2\right] \quad (18)
$$

$$
\le (1 - \nu\beta)^2\mathbb{E}\left[\|\nabla f_{\mathcal{S}_t}(\boldsymbol{x}_t) - \nabla f(\boldsymbol{x}_t)\|^2\right] + \mathbb{E}\left[\|\boldsymbol{d}_{t-1} - \nabla f(\boldsymbol{x}_t)\|^2\right]. \quad (19)
$$

On the other hand, Proposition A.1 guarantees that

$$
\mathbb{E}\left[\left\|\boldsymbol{\omega}_t^{\text{QHM}}\right\|^2\right] = (1 - \nu\beta)\mathbb{E}\left[\|\nabla f_{\mathcal{S}_t}(\boldsymbol{x}_t) - \nabla f(\boldsymbol{x}_t)\|^2\right] + \nu\beta\mathbb{E}\left[\|\boldsymbol{d}_{t-1} - \nabla f(\boldsymbol{x}_t)\|^2\right]
$$
$$
- \nu\beta(1 - \nu\beta)\mathbb{E}\left[\|\boldsymbol{d}_{t-1} - \nabla f_{\mathcal{S}_t}(\boldsymbol{x}_t)\|^2\right]. \tag{20}
$$

From (18) and (20), we have

$$
\mathbb{E}\left[\|\boldsymbol{d}_{t-1} - \nabla f(\boldsymbol{x}_t)\|^2\right] = \mathbb{E}\left[\|\boldsymbol{d}_{t-1} - \nabla f_{\mathcal{S}_t}(\boldsymbol{x}_t)\|^2\right] - \mathbb{E}\left[\|\nabla f_{\mathcal{S}_t}(\boldsymbol{x}_t) - \nabla f(\boldsymbol{x}_t)\|^2\right] \tag{21}
$$
$$
\leq \mathbb{E}\left[\|\boldsymbol{d}_{t-1} - \nabla f_{\mathcal{S}_t}(\boldsymbol{x}_t)\|^2\right]. \tag{22}
$$

Therefore, from (19) and (21), we obtain

$$
\mathbb{E}\left[\left\|\boldsymbol{\omega}_t^{\text{QHM}}\right\|^2\right] \leq \nu\beta(-2 + \nu\beta)\mathbb{E}\left[\|\nabla f_{\mathcal{S}_t}(\boldsymbol{x}_t) - \nabla f(\boldsymbol{x}_t)\|^2\right] + \mathbb{E}\left[\|\boldsymbol{d}_{t-1} - \nabla f_{\mathcal{S}_t}(\boldsymbol{x}_t)\|^2\right]. \tag{23}
$$

Then, let us show that, for all $t \in \mathbb{N}$,

$$
\mathbb{E}\left[\|\boldsymbol{d}_{t-1} - \nabla f_{\mathcal{S}_t}(\boldsymbol{x}_t)\|^2\right] \leq \nu\beta(2 - \nu\beta)\mathbb{E}\left[\|\nabla f_{\mathcal{S}_t}(\boldsymbol{x}_t) - \nabla f(\boldsymbol{x}_t)\|^2\right]. \tag{24}
$$

If (24) does not hold, there exists $t_0 \in \mathbb{N}$ such that

$$
\mathbb{E}\left[\left\|\boldsymbol{d}_{t_0-1} - \nabla f_{\mathcal{S}_{t_0}}(\boldsymbol{x}_{t_0})\right\|^2\right] > \nu\beta(2 - \nu\beta)\mathbb{E}\left[\left\|\nabla f_{\mathcal{S}_{t_0}}(\boldsymbol{x}_{t_0}) - \nabla f(\boldsymbol{x}_{t_0})\right\|^2\right],
$$

which implies

$$
\mathbb{E}\left[\left\|\nabla f_{\mathcal{S}_{t_0}}(\boldsymbol{x}_{t_0}) - \nabla f(\boldsymbol{x}_{t_0})\right\|^2\right] < \frac{1}{\nu\beta(2 - \nu\beta)}\mathbb{E}\left[\left\|\boldsymbol{d}_{t_0-1} - \nabla f_{\mathcal{S}_{t_0}}(\boldsymbol{x}_{t_0})\right\|^2\right]. \tag{25}
$$

Hence, from (23) and (25),

$$
\mathbb{E}\left[\left\|\boldsymbol{\omega}_{t_0}^{\text{QHM}}\right\|^2\right] < \nu\beta(-2 + \nu\beta)\left\{\frac{1}{\nu\beta(2 - \nu\beta)}\mathbb{E}\left[\left\|\boldsymbol{d}_{t_0-1} - \nabla f_{\mathcal{S}_{t_0}}(\boldsymbol{x}_{t_0})\right\|^2\right]\right\}
$$
$$
+ \mathbb{E}\left[\left\|\boldsymbol{d}_{t_0-1} - \nabla f_{\mathcal{S}_{t_0}}(\boldsymbol{x}_{t_0})\right\|^2\right]
$$
$$
= 0.
$$

Since $\mathbb{E}\left[\left\|\boldsymbol{\omega}_{t_0}^{\text{QHM}}\right\|^2\right] \geq 0$, there is a contradiction. Therefore, (24) holds for all $t \in \mathbb{N}$. Then, Lemma A.1, (18), (22), and (24) ensure that

$$
\mathbb{E}\left[\left\|\boldsymbol{\omega}_t^{\text{QHM}}\right\|^2\right] \leq (1 - \nu\beta)^2\mathbb{E}\left[\|\nabla f_{\mathcal{S}_t}(\boldsymbol{x}_t) - \nabla f(\boldsymbol{x}_t)\|^2\right]
$$
$$
+ \nu^3\beta^3(2 - \nu\beta)\mathbb{E}\left[\|\nabla f_{\mathcal{S}_t}(\boldsymbol{x}_t) - \nabla f(\boldsymbol{x}_t)\|^2\right]
$$
$$
= \left\{(1 - \nu\beta)^2 + \nu^3\beta^3(2 - \nu\beta)\right\}\mathbb{E}\left[\|\nabla f_{\mathcal{S}_t}(\boldsymbol{x}_t) - \nabla f(\boldsymbol{x}_t)\|^2\right]
$$
$$
\leq \frac{1}{1 - \nu\beta}\frac{C_{\text{QHM}}^2}{b}.
$$

This completes the proof. □

### C.3 DERIVATION OF EQUATION (2)

Let $\boldsymbol{y}_t$ be the parameter updated by the gradient descent and $\boldsymbol{x}_{t+1}$ be the parameter updated by SHB at time $t$; i.e.,

$$
\boldsymbol{y}_t := \boldsymbol{x}_t - \eta\nabla f(\boldsymbol{x}_t),
$$
$$
\boldsymbol{x}_{t+1} := \boldsymbol{x}_t - \eta\boldsymbol{m}_t
$$
$$
= \boldsymbol{x}_t - \eta(\nabla f(\boldsymbol{x}_t) + \boldsymbol{\omega}_t^{\text{SHB}}).
$$

Then, we obtain

$$
\begin{aligned}
\boldsymbol{x}_{t+1} &:= \boldsymbol{x}_t - \eta \boldsymbol{m}_t \\
&= (\boldsymbol{y}_t + \eta \nabla f(\boldsymbol{x}_t)) - \eta \boldsymbol{m}_t \\
&= \boldsymbol{y}_t - \eta \boldsymbol{\omega}_t^{\text{SHB}},
\end{aligned}
\tag{26}
$$

from $\boldsymbol{\omega}_t^{\text{SHB}} := \boldsymbol{m}_t - \nabla f(\boldsymbol{x}_t)$. Hence,

$$
\begin{aligned}
\boldsymbol{y}_{t+1} &= \boldsymbol{x}_{t+1} - \eta \nabla f(\boldsymbol{x}_{t+1}) \\
&= \boldsymbol{y}_t - \eta \boldsymbol{\omega}_t^{\text{SHB}} - \eta \nabla f(\boldsymbol{y}_t - \eta \boldsymbol{\omega}_t^{\text{SHB}}).
\end{aligned}
$$

By taking the expectation with respect to $\boldsymbol{\omega}_t^{\text{SHB}}$ on both sides, we obtain, from $\mathbb{E}_{\boldsymbol{\omega}_t^{\text{SHB}}}\left[\boldsymbol{\omega}_t^{\text{SHB}}\right] = \boldsymbol{0}$,

$$
\mathbb{E}_{\boldsymbol{\omega}_t^{\text{SHB}}}\left[\boldsymbol{y}_{t+1}\right] = \mathbb{E}_{\boldsymbol{\omega}_t^{\text{SHB}}[\boldsymbol{y}_t]} - \eta \nabla \mathbb{E}_{\boldsymbol{\omega}_t^{\text{SHB}}}\left[f(\boldsymbol{y}_t - \eta \boldsymbol{\omega}_t^{\text{SHB}})\right],
$$

where we have used $\mathbb{E}_{\boldsymbol{\omega}_t}\left[\nabla f(\boldsymbol{y}_t - \eta \boldsymbol{\omega}_t)\right] = \nabla \mathbb{E}_{\boldsymbol{\omega}_t}\left[f(\boldsymbol{y}_t - \eta \boldsymbol{\omega}_t)\right]$, which holds for the Lipschitz-continuous and the differentiability of $f$ (Shapiro et al., 2009, Theorem 7.49). These conditions are guaranteed in our Assumption (A1). In addition, from (26) and $\mathbb{E}_{\boldsymbol{\omega}_t^{\text{SHB}}}\left[\boldsymbol{\omega}_t^{\text{SHB}}\right] = \boldsymbol{0}$, we obtain

$$
\mathbb{E}_{\boldsymbol{\omega}_t^{\text{SHB}}}\left[\boldsymbol{x}_{t+1}\right] = \boldsymbol{y}_t.
$$

Therefore, on average, parameter $\boldsymbol{x}_{t+1}$ of function $f$ arrived at using the SHB method coincides with parameter $\boldsymbol{y}_t$ of smoothed function $\hat{f}(\boldsymbol{y}_t) := \mathbb{E}_{\boldsymbol{\omega}_t^{\text{SHB}}}\left[f(\boldsymbol{y}_t - \eta \boldsymbol{\omega}_t^{\text{SHB}})\right]$ arrived at using gradient descent. A similar discussion yields a similar equation for QHM.

### C.4 Details of calculating degree of smoothing in Figure 3

**Training ResNet18 on CIFAR100 dataset:** From (3)-(5), the hyperparameters used in the experiments, and the value estimated in Section 4.3 for training ResNet18 on the CIFAR100 dataset, the degree of smoothing can be calculated as

$$
\delta^{\text{SGD}} = \eta \sqrt{\frac{C_{\text{SGD}}^2}{b}} = 0.1 \cdot \sqrt{\frac{1280}{b}} = \sqrt{\frac{12.8}{b}},
$$

$$
\begin{aligned}
\delta^{\text{SHB}} &= \eta \sqrt{\frac{C_{\text{SHB}}^2}{b} + \frac{\beta(\beta^2 - \beta + 1)}{(1-\beta)^2}\left(\frac{C_{\text{SHB}}^2}{b} + K_{\text{SHB}}^2\right)} \\
&= 0.1 \cdot \sqrt{\frac{25.318}{b} + \frac{0.9 \cdot 0.91}{(0.1)^2}\left(\frac{25.318}{b} + (1.77)^2\right)} \\
&= 0.1 \cdot \sqrt{82.9 \cdot \frac{25.318}{b} + 81.9 \cdot 3.1329} \\
&\approx \sqrt{\frac{21}{b} + 2.57},
\end{aligned}
$$

$$
\begin{aligned}
\delta^{\text{NSHB}} &= \eta \sqrt{\frac{1}{1-\beta} \cdot \frac{C_{\text{NSHB}}^2}{b}} = 0.1 \cdot \sqrt{\frac{1}{1-0.9} \cdot \frac{128}{b}} \\
&= 0.1 \cdot \sqrt{10 \cdot \frac{128}{b}} \\
&\approx \sqrt{\frac{12.8}{b}},
\end{aligned}
$$

where $\eta = 0.1$ and $\beta = 0.9$ were used in the experiments, $C_{\text{SGD}}^2 = 1280, C_{\text{SHB}}^2 = 25.318$, and $C_{\text{NSHB}}^2 = 128$ were calculated in Section 4.3, and $K_{\text{SHB}} := 1.77$ was observed in Section B.5. Figure 14 plots the computed degrees of smoothing $\delta^{\text{SGD}}, \delta^{\text{SHB}}$, and $\delta^{\text{NSHB}}$ versus batch size $b$ in training ResNet18 on CIFAR100. Figure 15 is a logarithmic graph version of Figure 14.

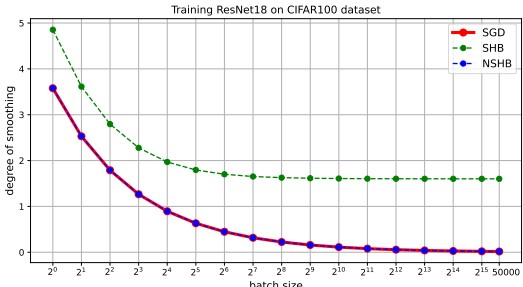 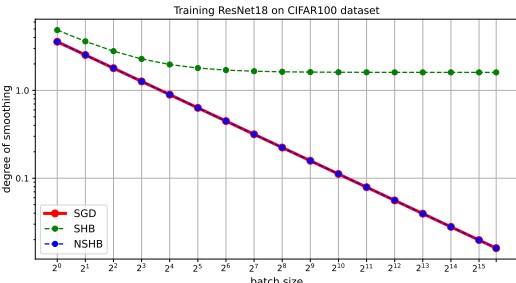

Figure 14: Degrees of smoothing $\delta^{\mathrm{SGD}}, \delta^{\mathrm{SHB}}$, and $\delta^{\mathrm{NSHB}}$ versus batch size in training ResNet18 on CIFAR100 dataset. **This is the same graph shown in Figure 3.**

Figure 15: Logarithmic graph version of Figure 14, clearly showing that $\delta^{\mathrm{SGD}}$ becomes smaller as the batch size is increased.

**Training WideResNet-28-10 on CIFAR100 dataset:** A similar argument can be made for the WideResNet-28-10 training. From (3)-(5), the hyperparameters used in the experiments, and the value estimated in Section 4.3 for training WideResNet-28-10 on the CIFAR100 dataset, the degree of smoothing can be calculated as

$$\delta^{\mathrm{SGD}} = \eta\sqrt{\frac{C_{\mathrm{SGD}}^2}{b}} = 0.1 \cdot \sqrt{\frac{10}{b}} = \sqrt{\frac{0.1}{b}},$$

$$\begin{aligned}
\delta^{\mathrm{SHB}} &= \eta\sqrt{\frac{C_{\mathrm{SHB}}^2}{b} + \frac{\beta(\beta^2 - \beta + 1)}{(1-\beta)^2}\left(\frac{C_{\mathrm{SHB}}^2}{b} + K_{\mathrm{SHB}}^2\right)} \\
&= 0.1 \cdot \sqrt{\frac{0.79}{b} + \frac{0.9 \cdot 0.91}{(0.1)^2}\left(\frac{0.79}{b} + (1.66)^2\right)} \\
&= 0.1 \cdot \sqrt{82.9 \cdot \frac{0.79}{b} + 81.9 \cdot 2.7556} \\
&\approx \sqrt{\frac{0.65}{b} + 2.26},
\end{aligned}$$

$$\begin{aligned}
\delta^{\mathrm{NSHB}} &= \eta\sqrt{\frac{1}{1-\beta} \cdot \frac{C_{\mathrm{NSHB}}^2}{b}} = 0.1 \cdot \sqrt{\frac{1}{1-0.9} \cdot \frac{1}{b}} \\
&= 0.1 \cdot \sqrt{10 \cdot \frac{1}{b}} \\
&\approx \sqrt{\frac{0.1}{b}},
\end{aligned}$$

where $\eta = 0.1$ and $\beta = 0.9$ were used in the experiments, $C_{\mathrm{SGD}}^2 = 10, C_{\mathrm{SHB}}^2 = 0.79$, and $C_{\mathrm{NSHB}}^2 = 1.0$ were calculated in Section 4.3, and $K_{\mathrm{SHB}} := 1.66$ was observed in Section B.5.

Figure 18 plots the computed degrees of smoothing $\delta^{\mathrm{SGD}}, \delta^{\mathrm{SHB}}$, and $\delta^{\mathrm{NSHB}}$ versus batch size $b$ in training WideResNet-28-10 on CIFAR100. Figure 19 is a logarithmic graph version of Figure 18 showing that, for WideResNet-28-10 as well, the degree of smoothing with SGD with momentum is always greater than with SGD. A comparison of Figures 14 and 18 shows that each optimizer was more robust to batch size in training WideResNet-28-10 than in training ResNet18. Therefore, generalizability may be less affected by batch size for training WideResNet-28-10 than for training ResNet18. This is shown to be true in Appendix D.1.

A similar argument can be made for training MobileNetV2 on CIFAR100 dataset (Figures 16 and 17) and ResNet18 on CIFAR10 dataset (Figures 20 and 21).

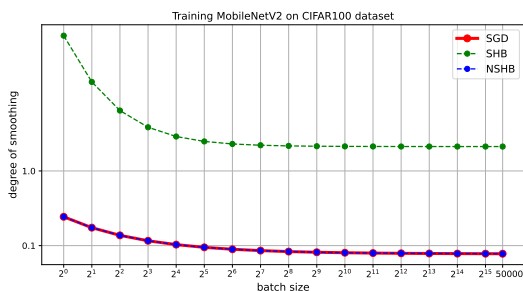
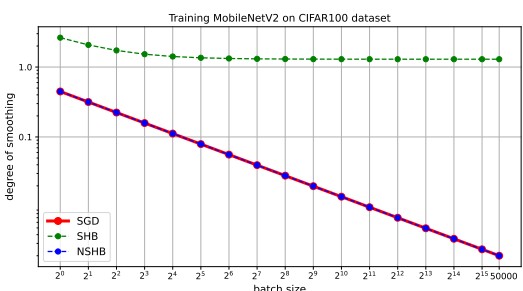

Figure 16: Degrees of smoothing $\delta^{\mathrm{SGD}}, \delta^{\mathrm{SHB}}$, and $\delta^{\mathrm{NSHB}}$ versus batch size in training MobileNetV2 on CIFAR100 dataset.

Figure 17: Logarithmic graph version of Figure 16 more clearly showing that $\delta^{\mathrm{SGD}}$ becomes smaller as the batch size is increased.

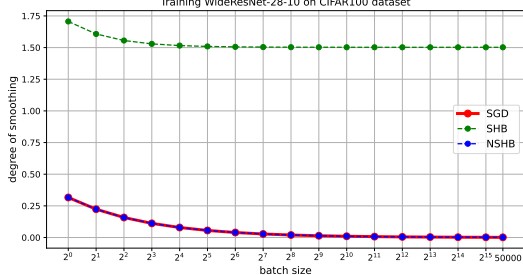
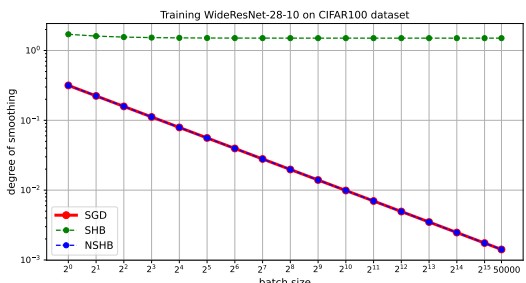

Figure 18: Degrees of smoothing $\delta^{\mathrm{SGD}}, \delta^{\mathrm{SHB}}$, and $\delta^{\mathrm{NSHB}}$ versus batch size in training WideResNet-28-10 on CIFAR100 dataset.

Figure 19: Logarithmic graph version of Figure 18 more clearly showing that $\delta^{\mathrm{SGD}}$ becomes smaller as the batch size is increased.

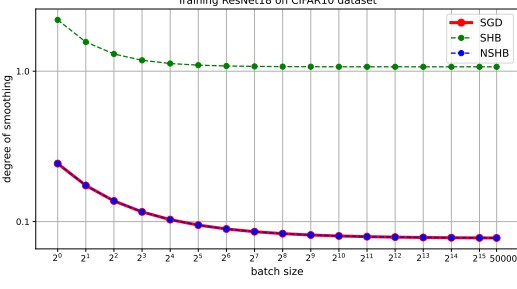
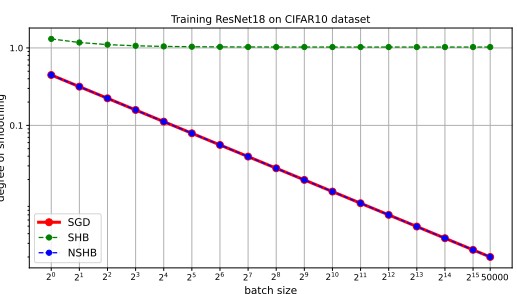

Figure 20: Degrees of smoothing $\delta^{\mathrm{SGD}}, \delta^{\mathrm{SHB}}$, and $\delta^{\mathrm{NSHB}}$ versus batch size in training ResNet18 on CIFAR10 dataset.

Figure 21: Logarithmic graph version of Figure 20 more clearly showing that $\delta^{\mathrm{SGD}}$ becomes smaller as the batch size is increased.

## D  MORE DETAILS ON EXPERIMENTAL RESULTS IN SECTION 5

This section complements Section 5. The experimental environment was as follows: NVIDIA GeForce RTX 4090×2GPU and Intel Core i9 13900KF CPU. The software was Python 3.10.12, PyTorch 2.1.0, and CUDA 12.2. The code is available at `https://anonymous.4open.science/r/role-of-momentum`.

### D.1  EXPERIMENTS ON GENERALIZABILITY OF SHB AND NSHB

We suggest that the generalizability of the model is determined by the degree of smoothing. In both SGD and SGD with momentum, if the degree of smoothing $\delta$ is too low, the process can be considered equivalent to optimizing a function $\hat{f}_\delta$ close to the original multimodal function $f$ by gradient descent, which leads to a sharp local optimal solution and less than excellent generalizability. Therefore, a sufficiently large degree of smoothing is required to obtain sufficient generalizability. On the other hand, from Lemma 2.1, too high a degree of smoothing may conversely lead to large deviations from the original function and may prevent successful optimization. We confirmed these considerations by experiment. We used learning rate of $0.1$ and momentum factor of $0.9$ in all experiments.

As shown in Figures 3 and 18, the degree of smoothing with both SHB methods stopped decreasing and stagnated from a certain batch size. Let us call $\hat{b}_{\text{SHB}}$ the batch size at which stagnation begins. For the training of ResNet18, $\hat{b}_{\text{SHB}} = 2^7$, while for the training of WideResNet-28-10, $\hat{b}_{\text{SHB}} = 2^4$. Therefore, when using an SHB method with a batch size greater than $2^7$ for the training of ResNet18 and greater than $2^4$ for the training of WideResNet-28-10, the generalizability should be approximately equal since they can be regarded as optimizing smoothed functions with noise levels approximately equal.

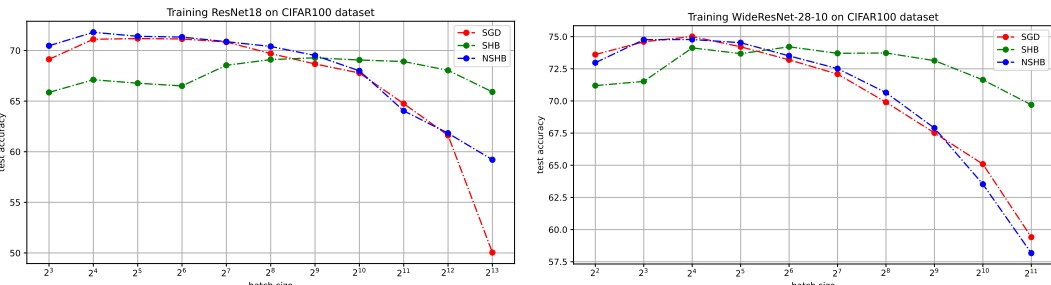

Figure 22: Test accuracy for SGD, SHB, and NSHB versus batch size in training ResNet18 on CIFAR100 dataset. **This is the same graph shown in Figure 4**.

Figure 23: Test accuracy for SGD, SHB, and NSHB versus batch size in training WideResNet-28-10 on CIFAR100 dataset.

We measured test accuracy with batch sizes of $2^3$ to $2^{13}$ for 200 epochs for training ResNet18 (Figure 22) and with batch sizes of $2^2$ to $2^{11}$ for 200 epochs for training WideResNet-28-10 (Figure 23) with SGD, SHB, and NSHB on the CIFAR100 dataset. In both cases, the generalizability of SGD worsened as the batch size was increased, whereas that of SHB remained stable. Moreover, SHB achieved almost equal test accuracy from batch sizes of $2^8$ to $2^{12}$ for ResNet18 and from batch sizes of $2^4$ to $2^9$ for WideResNet-28-10. For very large batch sizes, i.e., $2^{13}$ for ResNet18 and $2^{10}$ and $2^{11}$ for WideResNet-28-10, accuracy decreased even though the degree of smoothing was the same. Note that these results are for 200 epochs for all batch sizes, meaning that the number of steps may have been insufficient for the larger batch sizes. When using the CIFAR100 dataset and 200 training epochs, the number of parameter update steps is $1,250,000$ for a batch size of $2^3$ but only $1400$ for a batch size of $2^{13}$.

### D.2  DISTRIBUTION OF SEARCH DIRECTION NOISE

We collected 3000 each of search direction noise $\omega_t^{\text{SHB}}$ and $\omega_t^{\text{NSHB}}$ and tested whether each element follows a normal distribution. They were collected at the point where ResNet18 had been

trained on the CIFAR100 dataset (10,000 steps). ResNet18 has about 11M parameters, so $\omega_t^{\text{SHB}}$ and $\omega_t^{\text{NSHB}}$ form an 11M-dimensional vector. Figures 24 and 25 plot the results for the $\omega_t^{\text{SHB}}$ and $\omega_t^{\text{NSHB}}$ elements from dimension 0 to dimension 100,000. Figures 26 and 27 present the results for all elements. These results demonstrate that each search direction noise, $\omega_t^{\text{SHB}}$ and $\omega_t^{\text{NSHB}}$, follows a normal distribution.

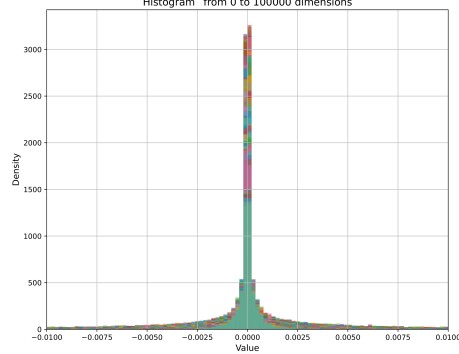 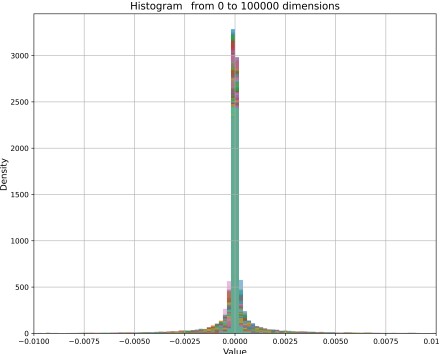

Figure 24: Distribution of 3000 $\omega_t^{\text{SHB}}$ elements from 0 to 100,000 dimensions.

Figure 25: Distribution of 3000 $\omega_t^{\text{NSHB}}$ elements from 0 to 100,000 dimensions.

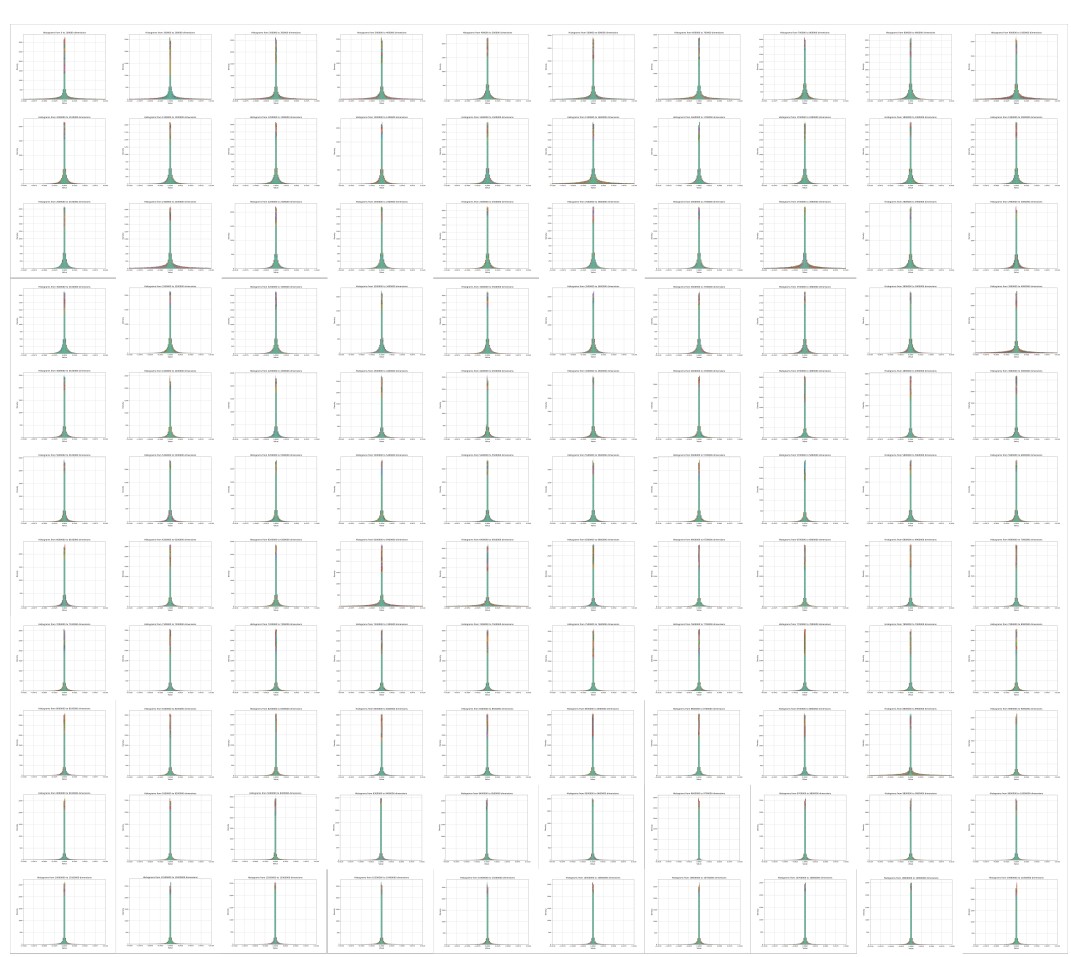

Figure 26: Complete results for distribution of 3000 $\omega_t^{\text{SHB}}$ elements. The distribution is plotted separately for each 100,000 dimensions.

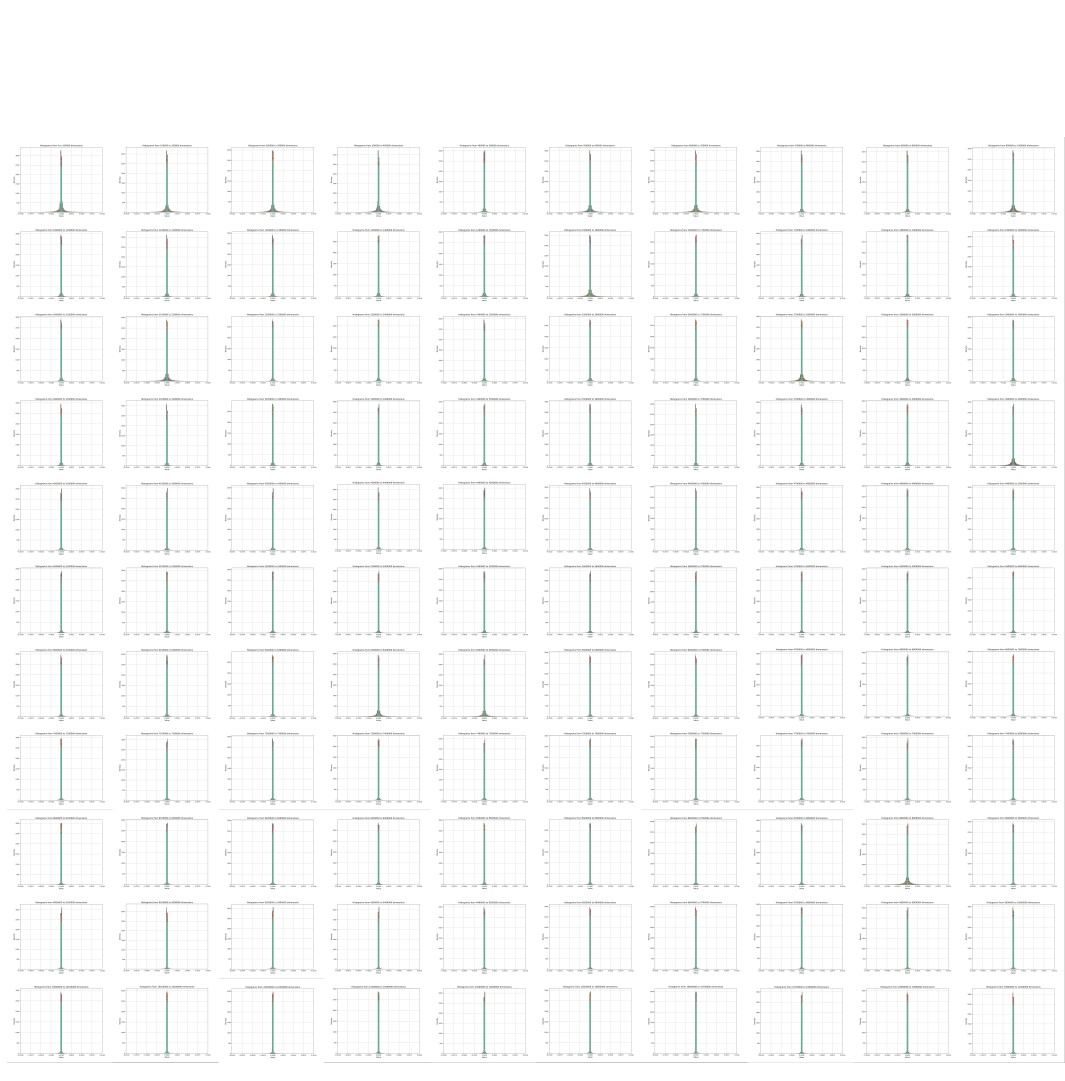

Figure 27: Complete results for distribution of 3000 $\omega_t^{\text{NSHB}}$ elements. The distribution is plotted separately for each 100,000 dimensions.

