# OpenReview forum: "Role of Momentum in Smoothing Objective Function and Generalizability of Deep Neural Networks"
_ICLR.cc/2025/Conference — ICLR 2025 Conference Withdrawn Submission_

### Official Review · Reviewer_Nrrc · 2024-10-17

**Soundness:** 1
**Presentation:** 2
**Contribution:** 2
**Rating:** 3
**Confidence:** 4

**Summary:**

This paper focus on the role of momentum for SGD. The authors explain theoretically the improvement induce by some momentum strategies, including SHB and QHM. They argue that momentum induce a degree of smoothness, that they quantify, on the objective function. This lead to an explanation for the better generalizability of SHB compare to SGD. In fact, flat critical point generalize better. Finally, they provide numerical experiments on CIFAR100 and CIFAR10 image classification to support their theoretical claims.

**Strengths:**

- The paper gives a good introduction to the problem of understanding momentum in a stochastic context.
- The interesting notion of "search direction noise" is introduces for the analysis, instead of the stochastic noise of the gradient estimator.
- Experiments are run in realistic context with neural network image classification.

**Weaknesses:**

Major weaknesses:
- In line 255, the authors claims that $\omega_t^{SHB}$ is Gaussian. However, in the proof of Theorem 3.1 (Appendix C.2), no proof of this fact is given. This Gaussian argument is studied for a pratical point of view in Appendix D.2. This statement is supported only by histograms, no statistical tests are provided with histograms, it is hard for me to deduce that these histograms are sample from a Gaussian distribution. Unfortunately, this Gaussian argument is the key point of the paper. In fact the degree of smoothness relies on this Gaussian property.
- Assumption (A1) is not verify for a large class of function including quadratic function, which is comonly used in machine learning. Therefore Theorem 4.1, 4.2, Proposition 4.1 do not hold for quadratic objective function.

Minor weaknesses:
- The paper has many typos and over-complex formulation such as:
	- line 85 : the sentence "This paper focuses on SHB and QHM, which covers many momentum methods, especially NSHB, but does not cover SHB." is confusing for me.
	- line 184 : "let $G_{\xi_t}(x)$ be the stochatsic gradient of $f$ at $x$", do you mean that $G_{\xi_t}(x)$ is an unbiased stochastic estimation of the gradient $\nabla f$ ?
	- In Lemma 2.1, the function $f$ is supposed to be $L_f$ Lipschitz, this is not clearly state.
	- Assumption 2.1 (A1) Is it $f$ or $f_i$ that is $L_f$ Lipschitz ?
	- $\mathcal{S}$ has not been introduced before Assumption 2.1 (A3). Is it the set of all data points $z_i$ ?
	- line 267 : There is a $\eta$ missing in the expression, it must be $f(y_t - \eta \phi^{SHB} u_t)$.
	- Assumption 4.1 is equivalent to suppose that the sequence $x_t$ is bounded. If Assumption 4.1 is verify then the sequence is bounded $\|x_t\| \le D(0)$. For instance, in Theorem 4.1 of [1], the authors directly suppose that the sequence is bounded. I suggest to simplify this assumption into "the sequence $x_t$ is bounded", which is a common assumption in stochastic gradient descent algorithms analysis.
	- line 487 "there is an impressive correlation between the degree of smoothness and model generalizability" : it seems to not be a mathematical correlation but a more complex relation between these two quantities. In fact, if the degree of smoothness is too small or too large, model generalizability is low.
	- line 1578 : there is a typo in the latex compilation.

[1] Diederik P Kingma and Jimmy Lei Ba. Adam: A method for stochastic optimization. In Proceedings of the 3rd International Conference on Learning Representations, pp.1–15, 2015.

**Questions:**

- In line 1786, it is states that "These results demonstrate that each search direction noise, follows a normal distribution.". Can you run statistical test, such as Kolmogorov-Smirnov test, for this fact ?
- Is the Gaussian argument in line 255 an hypothesis of the paper or a demonstrated fact ?
- Can you precise why increasing the degree of smoothness will lead to a flatter landscape around local minimum (which lead to higher generalizability) ?
- Can you justify why Assumption (A1) is realistic ?
- The assumption 2.1 (A2) (ii), is-it realistic to have a constant independent of $x_t$ ? For my point of view, Figure 5 does not permit to conclude that the assumption is realistic. In fact, for a finite number of iteration, this assumption will be verify but for a non-defined number of iterations, it is not clear.
- Assumption A1 seems to imply A4, can you discuss this implication ?
- Can you give me an interpretation for Assumption 2.1 (A4) ?
- Line 273 : the process is a "gradient descent in sense of expectation", what do you mean exactly ? Equation (2) is not exactly a gradient descent, the argument of $\hat f_{\eta \psi^{SHB}}$ is $y_t$ and not $\mathbb{E}_{\omega_t^{SHB}}(y_t)$.
- Why there are two parameters $\nu$ and $\beta$ in Algorithm 2 ? For me since $\nu > 0$, it is a NSHB Algorithm with parameter $\nu \beta$.
- Theorem 4.1 and 4.2 are named convergence analysis but they do not prove that the sequence converges (even in a weak sens) to a local minimum or a critical point of $f$. Moreover Proposition A.2 suggests to minimize by $0$, $<x_t-x, \nabla f(x_t)>$ and not maximize it as in Theorem 4.1 and 4.2. Can you provide more explanation about the exact implication of these results ? Have this kind of convergence formulation been used in previous works ?

---

### Official Review · Reviewer_dWGD · 2024-10-18

**Soundness:** 2
**Presentation:** 2
**Contribution:** 1
**Rating:** 3
**Confidence:** 4

**Summary:**

This paper studies the role of momentum in smoothing objective functions from a gradient noise perspective and how it affects generalizability of DNNs. First, it analyzed the ``degree of smoothing’’. Second, it estimated the crtical batch size and the variance of stochastic gradients. Third, it empiricallu studied how the generalization depend on the degree of smoothing. Finally, it discussed the reported results and explained ``the contradiction that exists between momentum and stochastic noise’’.

**Strengths:**

-Inspired by random smoothing, this work proposed a novel concept ``degree of smoothing’’ which are connected to gradient noise and generalization. This perspective is reasonable to me.

-The ``degree of smoothing’’ can help theoretically analyze SGD, SHB, and QHM. The theoretical results are meaningful.

-Deriving the critical batch size is interesting.

**Weaknesses:**

-While the theoretical results are meaningful for gradient noise/variance analysis, they do not directly capture the connection to the generalization.

-The empirical evidences are not comprehensive enough. The experimental results only suggest the connection between generalization and the degree of smooth. The connections between generalization and some statistic are very common. I cannot how the ``degree of smooth’’ can predict generalization.

-SGD in PyTorch is SHB rather than NSHB. Line 80 made a false statement.

-The work needs to significantly improve literature review, including recent works on momentum, gradient noise & generalization.

-This work tends to make overclaim and ignore the contributions of a lot previous relevant works. The missed references including (but not limited to): [1] studied the convergence of momentum; [2] studied gradient noise/generalization of momentum; and a lot of studies analyze how gradient noise affect minima sharpness/generalization. Please carefully survey the papers on momentum, gradient noise & generalization.

Refs:

[1] Yan, Y., Yang, T., Li, Z., Lin, Q., & Yang, Y. (2018, July). A unified analysis of stochastic momentum methods for deep learning. In Proceedings of the 27th International Joint Conference on Artificial Intelligence (pp. 2955-2961).

[2] Xie, Z., Yuan, L., Zhu, Z., & Sugiyama, M. (2021, July). Positive-negative momentum: Manipulating stochastic gradient noise to improve generalization. In International Conference on Machine Learning (pp. 11448-11458). PMLR.

**Questions:**

Please see the weaknesses

---

### Official Review · Reviewer_T37s · 2024-11-04

**Soundness:** 2
**Presentation:** 3
**Contribution:** 2
**Rating:** 3
**Confidence:** 4

**Summary:**

The paper introduces a measure of smoothing effect for momentum methods and shows its impact on optimization and generalization. The main comparison is between three optimizers SGD, SHB and NSHB.

**Strengths:**

The paper provides clear images and illustrations for their experimental results.

**Weaknesses:**

The theory in this paper is flawed. This includes

1. The claim on line 254 is questionable, and as a result the viewpoint of function smoothing is invalid. The search direction differences $\omega_t^{SHB}$, $\omega_t^{QHM}$ are not Guassians in general. Even if the paper claims to have identified that their probability density looks like a gaussian curve, they may not be isotropic and its shape is crucial to the optimization and generalization performance of SGD and SHB (e.g. https://arxiv.org/abs/1803.00195). Even if it is isotropic gaussian, line 254 may also be incorrect as Theorem 3.1 only gives upper bounds of the Guassian norm, so one cannot establish equality in line 254.

2. From figure 1 and line 282-285 the authors depict a different trend of smoothing change across batch sizes between SHB and QHM (or NSHB), which is inaccurate. In figure 1 the trend is only given by the upper bound estimate (3)-(5) and the difference in trend may be only due to the fact that (4) and (5) use different ways to estimate the upper bounds, which are not tight. For instance at an infinity batch size, as there is no noise at all in all the gradients, there should not be any noise in the SHB updates. Actually QHM is only a reparameterization of SHB so their degree of smoothing should follow similar trends. This point is also corroborated by Theorem 4.1 & 4.2 that the bounds for QHM and SHB are similar and the phenomenon in figure 1 is not observed here.

It is also confusing to the readers that the smoothing measures seem to have no connection to the result of theorem 4.1&4.2, and the results of theorem 4.1&4.2 seem to have little connections to the actual optimization and generalization behavior in the general non-convex optimization regime.

3. It has been established by previous work (e.g. https://arxiv.org/abs/2307.15196) that the degree of smoothing is not responsible for explaining generalization for momentum, which is contradictory to the claim of the current paper. The experimental results in this paper are well-observed by previous works that QHM and SGD have similar behaviors; and the behavior difference QHM and SHB is due to a change of effective learning rate. By setting SHB to have learning rate $\eta(1-\beta)$, previous work also shows that the new SHB and SGD have similar performances with different degrees of smoothing.

**Questions:**

Can you provide more clarification to the above weaknesses?

---

### Official Review · Reviewer_GN94 · 2024-11-07

**Soundness:** 3
**Presentation:** 3
**Contribution:** 3
**Rating:** 6
**Confidence:** 4

**Summary:**

This paper takes on an interesting challenge by exploring the role of momentum in SGD and its effect on both smoothing and generalizability. The authors consider the question: if momentum reduces gradient noise, how does it still end up improving generalizability? By leveraging the idea of search direction noise—a unique take on the noise added by momentum—the authors offer a plausible explanation that ties smoothing directly to the model’s generalization performance. Their approach combines theoretical insights with experiments, primarily focused on SHB (Stochastic Heavy Ball) and QHM (Quasi-Hyperbolic Momentum) variants.

**Strengths:**

- The concept of search direction noise is novel. By framing momentum’s effects through this new type of noise, the authors provide a clearer understanding of how momentum contributes to generalization.
- The theoretical analysis is solid, and the empirical results back it up nicely. The authors link specific hyperparameters to the degree of smoothing, which should make these insights practically valuable for tuning models.
- This paper is well written, with a clear flow of ideas.

**Weaknesses:**

- In section 4.2, the authors use the critical batch size to estimate the variance of the stochastic gradient of certain optimizer, which seems quite strange to me. Can't we just store the iterates $\lbrace x_0, \ldots, x_t \rbrace$, and the batches to directly compute the variance, which should be much more accurate? Using this ground truth variance would verify the connection between critical batch size and variance.
- While the ResNet18 and CIFAR100 experiments are useful, they feel a bit narrow. Adding experiments on a variety of architectures, like Transformers or larger datasets (e.g., ImageNet), would help make the conclusions feel more robust and applicable across different tasks.

**Questions:**

- I am also curious about Nesterov's version of momentum, which is also covered by the QHM formulation if I remember correctly. Does it show similar behavior as SHB in terms of the degree of smoothing?
- I am curious about whether there exists a method that can somehow adapt to the degree of smoothing, similar to sharpness-aware minimization (SAM) that adapts to the sharpness.

---

### Note · Authors · 2024-12-04

**Comment:**

Extensive revisions are necessary to address the reviewers' comments. The paper will be withdrawn.
We sincerely appreciate the hard work of the reviewers.

**Withdrawal Confirmation:**

I have read and agree with the venue's withdrawal policy on behalf of myself and my co-authors.